

# 1 Photochemical and ozone-induced aging significantly alter the
# 2 viscosity of aqueous *trans*-aconitic acid aerosol particles

Cynthia Antossian[1], Marcel Müller[1,a], and Ulrich K. Krieger[1]
[1]Institute for Atmospheric and Climate Science, ETH Zurich, Universitätstrasse 16, 8092 Zurich, Switzerland
[a]currently at: Institute of Biogeochemistry and Pollutant Dynamics, ETH Zurich, Universitätstrasse 16, 8092
Zurich, Switzerland
*Correspondence to*: Ulrich K. Krieger (ulrich.krieger@env.ethz.ch) and Cynthia Antossian
(cynthia.antossian@env.ethz.ch)
**Abstract.** Aging processes of organic aerosols, including reactions with gas phase oxidants, such as ozone ($O_3$),
as well as photochemical reactions, can significantly alter their physicochemical properties. While previous
research has examined how photochemical aging and ozonolysis affect the physicochemical properties of organic
aerosols, our study investigates the combined effect of photolysis and ozonolysis. We use aqueous *trans*-aconitic
acid as a proxy for secondary organic aerosol particles (SOA), selected for its ability to absorb UV light and for
containing a C=C double bond that is susceptible to ozonolysis. We observe significant mass loss in single particles
levitated in an electrodynamic balance when exposed to either $O_3$ or UV light (375 nm), as well as to both aging
processes simultaneously, resulting from fragmentation reactions followed by the volatilization of some of the
products. Viscosity measurements at 17% relative humidity revealed an increase of nearly 4 orders of magnitude
after both UV exposure and combined UV and $O_3$ exposure at 60% mass loss. Interestingly, continued UV-aging
beyond 60% mass loss resulted in a viscosity decrease, whereas combined UV and $O_3$ exposure led to a further
viscosity increase. Hygroscopicity exhibited only a modest decline after 20% mass loss during UV-aging and
remained constant with further UV exposure; this reduction was less pronounced when UV-aging occurred in the
presence of $O_3$. Overall, our results indicate that the mixing times within accumulation mode SOA particles may
increase from 4 s to 4 h after aging under dry boundary layer conditions.

## 26 1 Introduction

Aerosol particles play a prominent role in the atmosphere, since they are involved in many important processes
including cloud formation, biogeochemical cycling, and light scattering (Seinfeld and Pandis, 2016). Moreover,
aerosols affect human health, since they have been associated with respiratory and cardiac diseases, oxidative
stress, and cancer (Dockery and Pope, 1994; Nel, 2005). Organic compounds constitute a major fraction of
atmospheric aerosols (20-90%) (Kanakidou et al., 2005; Jimenez et al., 2009). A significant fraction of these
aerosols are secondary organic aerosols (SOA) (Zhang et al., 2007), which are formed when volatile organic
compounds oxidize and produce compounds with low volatility that partition from the gas phase into the particle
phase, eventually forming SOA (Pankow, 1994). In addition, these low volatility compounds can partition into
preexisting aerosols, leading to internal mixing of primary and secondary aerosol particles (Marcolli et al., 2004;
Marcolli and Krieger, 2020)
Throughout their lifetimes in the atmosphere, these organic aerosol particles undergo various aging processes. One
of these processes are reactions with gas phase oxidants, such as ozone ($O_3$), hydroxyl radicals (OH), and nitrate
radicals ($NO_3$). In addition to the reactions with oxidants, organic aerosols can undergo photochemical reactions
due to exposure to UV or even visible light, either directly or indirectly by photosensitizers, leading to additional
aging. Oxidation reactions can occur through different pathways (Kroll and Seinfeld, 2008), depending on the
chemical composition of the gas and particle phase and on the atmospheric conditions, such as temperature and
relative humidity (RH). For instance, fragmentation can take place because of carbon-carbon double bond
cleavage, leading to more volatile products that volatilize from the particle. Functionalization can also take place
because of the addition of polar functional groups, thus increasing oxygen-to-carbon ratio O:C and forming more
hygroscopic, higher molecular weight products that have low volatility. In addition, oligomerization or accretion
can occur through the association of small molecules, consequently forming products with similar O:C, but higher
number of carbon atoms that have low hygroscopicity, low volatility, and high viscosity (Kroll and Seinfeld, 2008).
Fragmentation reactions have been observed for a range of organic aerosol particles during photochemical aging
or when exposed to atmospheric oxidants (O'brien and Kroll, 2019; Sun and Smith, 2024; Dou et al., 2021; Kroll





et al., 2015). Moreover, oxidation of SOA has been shown to induce the formation of oligomers through accretion
reactions e.g. (Kalberer et al., 2004). At present, it remains uncertain which mechanisms dominate under different
aerosol compositions and atmospheric conditions, making it challenging to predict aerosol properties during and
after aging.
These aging processes can significantly alter the physicochemical properties of organic aerosols, such as the
viscosity and hygroscopicity (Hosny et al., 2016; Athanasiadis et al., 2016; Baboomian et al., 2022; Jimenez et al.,
2009; Kroll et al., 2011; Kroll et al., 2015). Viscosity of organic aerosols varies over a wide range, depending on
the chemical properties, such as molecular structure, functional groups, chain length, molecular weight, and carbon
oxidation state and on the environmental conditions, such as temperature and relative humidity (Gou et al., 2025).
Moreover, the viscosity of organic aerosols is an indication of the phase state, with aerosols having viscosity values
less than $10^2$ Pa s considered as being liquid, between $10^2$ and $10^{12}$ Pa s semi-solid, and greater than $10^{12}$ Pa s solid
(Shiraiwa et al., 2011; Koop et al., 2011).
Viscosity affects the condensed phase chemistry (Pöschl and Shiraiwa, 2015), gas-particle partitioning (Shiraiwa
et al., 2011), and the ability of the aerosol particle to act as ice nucleating particles (Wolf et al., 2020; Murray et
al., 2010), while hygroscopicity affects the ability of the particle to act as cloud condensation nuclei (Chan et al.,
2008), thereby affecting the Earth's energy budget and climate. Moreover, viscosity affects the ability of the
particle to react with other chemical species. Viscous aerosols can limit molecular motion due to slow diffusion of
oxidants, water, and organic compounds (Pöschl and Shiraiwa, 2015). This makes particulate air pollutants less
susceptible to degradation and increases their mixing times in the troposphere, thereby favoring their transport and
effecting human health (Mu et al., 2018; Bastelberger et al., 2017) . For instance, polycyclic aromatic hydrocarbons
(PAHs) that were coated with viscous organic aerosols, were shielded from oxidation, highlighting stronger long-
range transport and elevated lung cancer risk (Shrivastava et al., 2017).
Several studies have observed the change in viscosity of organic aerosols upon aging. For instance, studies on the
ozonolysis of oleic acid aerosols (Hosny et al., 2016) and squalene droplets (Athanasiadis et al., 2016) indicate a
significant increase in viscosity upon oxidation. SOA formed from ozonolysis of organic aerosols might also
undergo additional aging, for instance photochemically, altering further the viscosity of these particles. A recent
study on the photochemical aging of SOA generated from the ozonolysis of d-limonene and α-pinene shows an
increase of several orders of magnitude in viscosity after UV-aging, ultimately transforming into a glassy solid
state, especially at low temperatures and relative humidities (Baboomian et al., 2022). However, to the best of our
knowledge, it is not known how the aerosol properties, particularly viscosity and hygroscopicity, will change after
simultaneous exposure to UV light and ozone.
In this work, we use *trans*-aconitic acid particles as surrogate for SOA to investigate how the viscosity and
hygroscopicity are influenced by different degrees and different combinations of photochemical and ozone-
induced aging.

## 2 Materials, Instrumentation, and Method

### 2.1 Materials

*Trans*-aconitic acid (AA) is a naturally occurring tricarboxylic acid that is produced by several plants and
accumulates significantly in sugar cane and sweet sorghum (Bruni and Klasson, 2022). Since it is a highly oxidized
molecule having carboxyl functional groups, it has low vapor pressure, and is water soluble, we use it here as a
proxy system for SOA present in the atmosphere.

**Figure 1: The structure of *trans*-aconitic acid.**





The UV-visible absorption spectra (Fig. 2) obtained using a UV-Visible spectrophotometer (Varian Cary 100 Bio)
shows that concentrated AA solution (2.13 mol $L^{-1}$) absorbs even in the visible range of the solar spectrum, with
significant absorbance at 375 nm. AA's absorbance at the visible to near UV wavelengths could be attributed to
the conjugated structure of *trans*-aconitic acid, which makes it possible to undergo photochemical reaction at 375
nm illumination. Moreover, the presence of the double bond makes AA susceptible to ozonolysis. As a result of all
these properties, AA was chosen in this study as a suitable proxy system for SOA for studying synergistic effects
between photochemistry and ozonolysis.

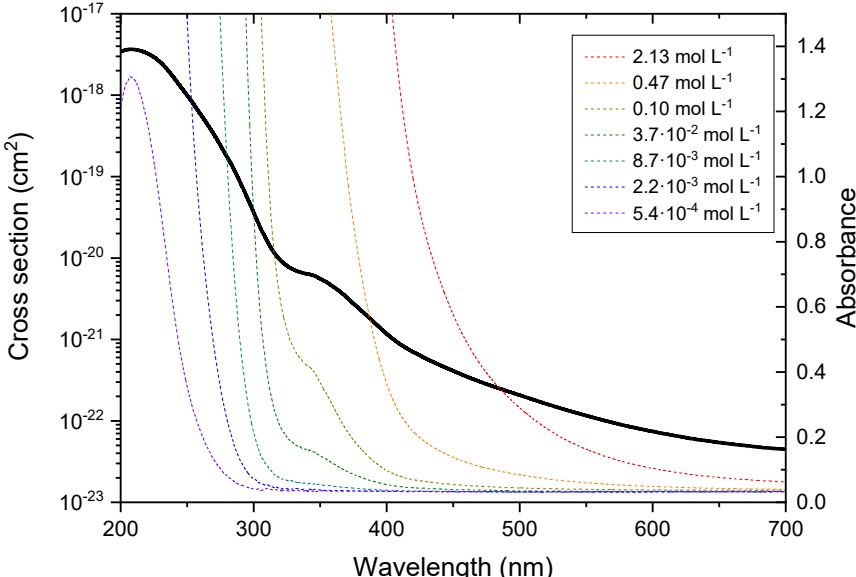


**Figure 2: Absorption spectra (colored lines) and cross section (black line) of AA solution of different concentrations as**
**a function of wavelength (absorbance values higher than 1.5 are not shown because they are above the maximum**
**detection limit of the instrument).**

**2.2 Electrodynamic Balance Setup**
Experiments on single, levitated, aqueous AA particles were conducted using a linear electrodynamic balance
(EDB) (Fig. 3). The set-up has been described previously (Müller et al., 2022). Briefly, single aqueous AA droplets
were prepared from 4 wt–% AA (98%, Sigma Aldrich) in MilliQ water and injected using a droplet on demand
generator (HP-inkjet cartridge model 51604). The droplet was inductively charged and levitated in the EDB trap
by AC and DC electric fields. 50 sccm of humidified flow from hydrocarbon free synthetic air (PanGas, 20% $O_2$
in $N_2$) was used to keep RH in the EDB at 78% – 88%. All experiments were performed at room temperature (~298
K). RH and temperature were measured using two sensors (Sensirion SHT85, Switzerland) placed at the flow
entrance to the EDB as well as at the flow exit of the EDB. The sensor at the flow exit is located close to the
levitated particle and it was confirmed in measurements of the hygroscopic response of NaCl particles that this
sensor's RH readout is close to the RH at the location of the particle. The uncertainty in RH measurement is ±
118   1.5%.

Aging experiments were conducted by exposing AA particles to 10 ppm ozone generated by UV photolysis of $O_2$
in synthetic air using an ozone generator (AOS, BMT Messtechnik, Germany). The readout of an electrochemical
ozone sensor (OX-B431 on Alphasense ISB, Alphasense, UK) was used to control the ozone generator. For UV



exposure, the unfocused beam of a 375 nm UV laser (Omicron LuxX-20, Germany) illuminated the particle
levitated at the second segment of the EDB. Its beam profile and power lead to an irradiance at the particle location
of about 0.16 W cm$^{-2}$.
The O$_3$ exposure used in the experiments is similar to an atmospheric exposure of 100 ppb ozone for 10 d. The
UV photon flux used in the experiments is 1 order of magnitude larger than atmospheric conditions, based on the
power density of the laser irradiation and AA absorbance together with the solar irradiance integrated over the
entire spectrum. This means that the O$_3$ exposure and UV irradiation exposure used in our experiments are
comparable to about 10 d of exposure in the atmosphere, which is a typical lifetime of atmospheric aerosols.
The particle was illuminated with a 532 nm diode pumped solid state laser (Thorlabs, DJ532, USA) and the
scattered light was collected on CCD cameras (Raspberry Pi HQ camera, UK). The change in mass of the particle
was deduced by measuring the electric force needed to balance the gravitational force of the particle. The DC
voltage was adjusted by an automatic feedback loop driven by the particle image. Since in addition to the
gravitational force, the DC voltage compensates for the Stokes force of the gas flow, drag force correction was
applied to deduce changes in mass. Moreover, the size of the particle was deduced from the two-dimensional
angular optical scattering (TAOS) pattern of laser-illuminated droplets. The inverse of the fringe distance along
the symmetry axis of the TAOS pattern is a direct measure of the size; smaller size changes are detected by
following a single maximum in the TAOS pattern. Thus, the radius of the particle was deduced from the mean
distance between the fringes as long as the particle remained spherical and symmetric with regular TAOS pattern.
For detecting phase transitions breaking the spherical symmetry of the particle, we use the pattern distortion
parameter as introduced by (Braun and Krieger, 2001). More details on the radius retrieval can be found in
Appendix A.

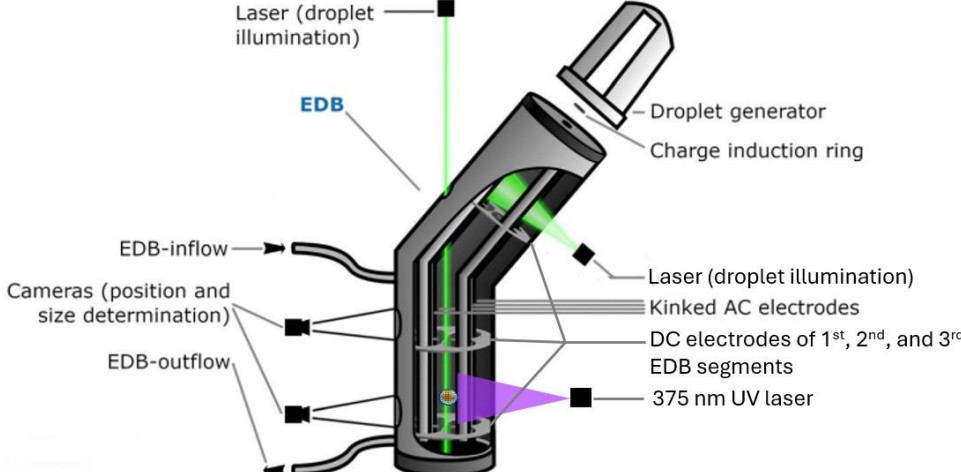


**Figure 3: Schematic representation of the EDB set-up, adopted with slight modifications from (Müller et al., 2022). The**
**UV laser illuminated the particle levitated at the 2$^{nd}$ segment of the EDB. For simplification and better representation,**
**the UV laser was shown in this figure in the 3$^{rd}$ segment of the EDB.**

### 2.3 Overview of the experimental procedure

Figure 4 shows an example of the experimental procedure to obtain hygroscopicity as well as viscosity for a
combined ozonolysis and photochemical aging experiment. Initially, an aqueous AA particle is injected and
levitated in the EDB. After RH is equilibrated at ~83%, 375 nm UV laser and 10 ppm O$_3$ are switched on (t=0).
The change in mass is monitored and size measurements are done in parallel. The reaction is carried out at
approximately constant RH, and UV illumination and O$_3$ are switched off after the particle loses 60% of its initial
mass. Since slight fluctuations in RH and temperature occur after switching off O$_3$, the particle is kept in the dark
until RH, and therefore the mass and size of the particle are almost constant. Kinetics of the reaction will be
discussed in Sect. 3.2. Afterwards, RH inside the EDB is decreased slowly at a rate of 0.2% min$^{-1}$ from ~83% to
~6% and then kept constant for 3.5 h to ensure sufficient drying. From this drying part of the experiment,




hygroscopicity is deduced, as will be explained in more detail in Sect. 2.4. Finally, RH is increased in two steps
rapidly, first from 6.3% to 28.2%, then from 28.2% to 49.4%, and finally raised back from 49.4% to 84.4% slowly.
While the flow change is indeed step-like, the response in RH is approximately exponential. The first step in
humidity was used to derive condensed phase water diffusivity, as will be discussed in more detail in Sect. 2.5.

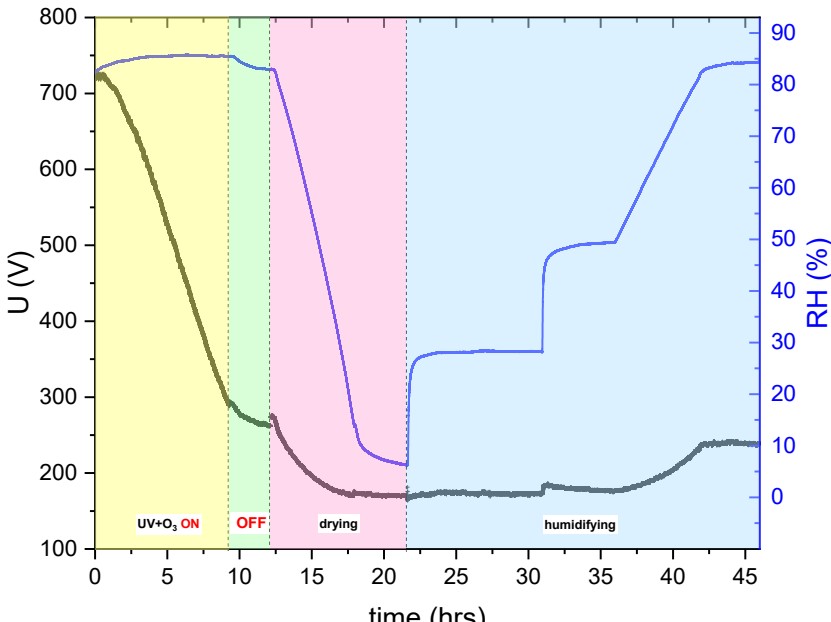


**Figure 4: Overview figure showing the complete experimental procedure of AA particle exposed to UV and O₃**
**simultaneously until 60% mass loss at room temperature. The black line represents the voltage needed to keep the**
**particle levitated in the EDB (corrected for drag force), which is proportional to the mass of the particle. The blue line**
**represents the RH data. The yellow section corresponds to switching on UV laser and O₃, the green section corresponds**
**to turning off UV and O₃, the pink section corresponds to drying, and the blue section corresponds to humidifying.**

**2.4 Hygroscopicity measurement**
Hygroscopicity and water diffusivity measurements were done on aged particles after they lost about 20%, 40%,
50%, 60% and 80% of their mass. For hygroscopicity measurements, RH was allowed to decrease slowly, at a rate
of 0.2% min$^{-1}$, by adjusting the ratio of dry nitrogen gas flow relative to the humidified flow while keeping the
total flow constant using automatic mass flow controllers. The change in mass of the particles was monitored
during drying using the automatic feedback loop, see Fig. 4.
The hygroscopic size growth factor ($G(RH)$) indicates the relative increase in size of the particles at a certain RH
relative to dry conditions in response to water uptake. If $G(RH)$ is measured at different RH, a single parameter,
$\kappa$, for hygroscopic size growth can be determined, assuming ideality (Petters and Kreidenweis, 2007):
$$G(RH) = \frac{D(RH)}{D_0} = \left(1 + \kappa \frac{a_w}{1-a_w}\right)^{1/3} \tag{1}$$
where $D(RH)$ is the mobility diameter of the particle at a certain RH, $D_0$ is the diameter of the particle at dry
conditions, and $a_w$ is the water activity, which is equal to RH at equilibrium.





Since here the mass growth factor ($g(RH)$) was measured instead of the size growth factor, $\kappa$ can be determined
from the mass growth factor by using Eq. (2) (Zardini et al., 2008):

$$g(RH) = \frac{m(RH)}{m_0} = 1 + ((G(RH))^3 - 1)\frac{\rho_w}{\rho_0} \tag{2}$$

Combining Eq. (1) and Eq. (2) yield Eq. 3:

$$g(RH) = 1 + \left(\kappa \frac{a_w}{1-a_w}\frac{\rho_w}{\rho_0}\right) \tag{3}$$

where $m(RH)$ is the mass of the particle at a certain RH, $m_0$ is the mass of the particle at dry conditions (measured
at ~6% RH), $\rho_w$ is the density of water, and $\rho_0$ is the density of the particle at dry conditions. $\rho_0$ for AA was
determined experimentally using the conventional additivity rule and was found to be 1.56 g cm⁻³ and was assumed
to remain constant after aging, see Appendix B.
Since most untreated AA effloresced at RH 59.9%, $\kappa$-parametrization fitting according to Eq. (3) was done only in
the RH range 62% – 80% for all particles. Furthermore, fitting at higher RH provides a better estimation for $\kappa$ as
deviations from ideality become less important with dilution. Figure 5 shows an example of the measured mass
growth together with a regression of Eq. (3) to deduce $\kappa$.

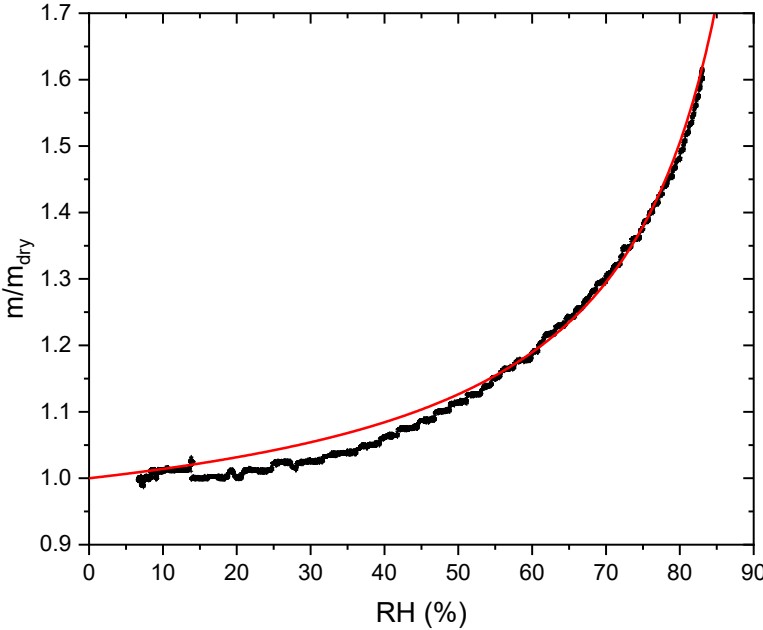


**Figure 5: Mass growth factor as a function of RH for AA exposed to UV light and O₃ till 60% mass loss (black line). The red line is the $\kappa$-parametrization fitting in the RH range 62% – 80%, yielding a $\kappa$ of 0.197.**


**2.5 Viscosity determination**

Viscosity of the particles after aging was inferred indirectly by estimating the water diffusivity and then using the
fractional Stokes–Einstein relation to deduce the viscosity.



For water diffusivity measurements, the aged particles were subjected to drying and were kept under low RH
(below 15%) for 4 ± 1 h, until RH stabilized around 6%. Then, RH was increased rapidly from 6.7 ± 0.7% to 28.6
± 0.5% and kept constant for 9 ± 2 h. As RH increases, water diffuses from the gas phase into the particle phase
until equilibrium between water activity throughout the particle and RH of the gas phase is reached (Koop et al.,
2011). The time taken to reach equilibrium depends on the viscosity of the particle, i.e. liquid droplets will reach
equilibrium faster than amorphous solid or glassy particles. Thus, water diffusivity can be inferred by observing
the time response in particle mass or size following a step increase in RH (Zobrist et al., 2011; Bones et al., 2012).
One way of retrieving the diffusion coefficient from such data is by using a numerical model, solving the diffusion
equation (Zobrist et al., 2011; O'meara et al., 2016). Alternatively, the response of step increase in RH may be
approximated by an exponential approach to the new RH setting, see data and exponential fit in Fig. 6. While a
viscous liquid close to the glass transition exhibits a non-linear response leading to a non-exponential relaxation
(Debenedetti and Stillinger, 2001; Rickards et al., 2015), a less viscous liquid may be reasonably well
approximated by a single exponential response. Under these conditions, the hygroscopic response can be
deconvoluted analytically as detailed in Appendix C. The response $y(t)$ to the hygroscopic 'step' in either mass or
size is then given by Eq. (4) with $\tau_1$ being the characteristic time of the response we are searching for and $\tau_2$ being
the characteristic time of the RH step, induced by the flow change from dry to humidified.
$$y(t) = \left(1 - e^{-t/\tau_1}\right) - \frac{\tau_2}{\tau_2 - \tau_1}\left(e^{-t/\tau_2} - e^{-t/\tau_1}\right) \tag{4}$$
Water diffusivity can then be approximately deduced from $\tau_1$ using the following equation (Seinfeld and Pandis,
2016; Bones et al., 2012):
$$\tau_1 = \frac{r^2}{\Pi^2 D} \tag{5}$$
where $\tau_1$ is the characteristic time, $r$ is the radius of the particle, and $D$ is the diffusion coefficient.
We show in Appendix D that this linear response approximation needs a correction to obtain $D$, such that it
compares favorably with a full numerical model for water diffusivities in the range of 1.7 x 10^-12 cm^2 s^-1 to 2x10^-9
cm^2 s^-1 and particle sizes used in our experiment.
Viscosity can then be predicted from diffusivity using Stokes–Einstein relation (Einstein, 1905):
$$D = \frac{KT}{6\Pi\eta R_{diff}} \tag{6}$$
where $D$ is the diffusion coefficient, $K$ is the Boltzmann constant, $T$ is the temperature, $\eta$ is the viscosity, and $R_{diff}$
is the radius of the diffusing species. While this relation works well for large diffusing molecules in which the
radius of the diffusing molecule ($R_{diff}$) is greater than that of the organic matrix molecule ($R_{matrix}$), it
underestimates the diffusion coefficients in organic–water mixtures by several orders of magnitude for small
diffusing molecules like water, where $R_{diff}$ is smaller than $R_{matrix}$ (Price et al., 2015; Bastelberger et al., 2017).
A better approach to relate diffusivity to viscosity is to use fractional Stokes-Einstein relation where $D$ is
proportional to $1/\eta^\xi$, $\xi$ being the fractional exponent, which depends on the ratio $R_{diff}/R_{matrix}$ (Evoy et al., 2020).
$\xi$ can be expressed in terms of $R_{diff}/R_{matrix}$ according to the following empirical equation (Evoy et al., 2020):
$$\xi = 1 - [0.73\exp{(-1.79\frac{R_{diff}}{R_{matrix}})}] \tag{7}$$
The hydrodynamic radius of AA ($R_{matrix}$) was assumed to be the same as that of citric acid, which is structurally
similar to aconitic acid, and was taken as 3.7 Å (Muller and Stokes, 1957) and for water the Van der Waals radius
of 1.41 Å (Pang, 2014) was used.



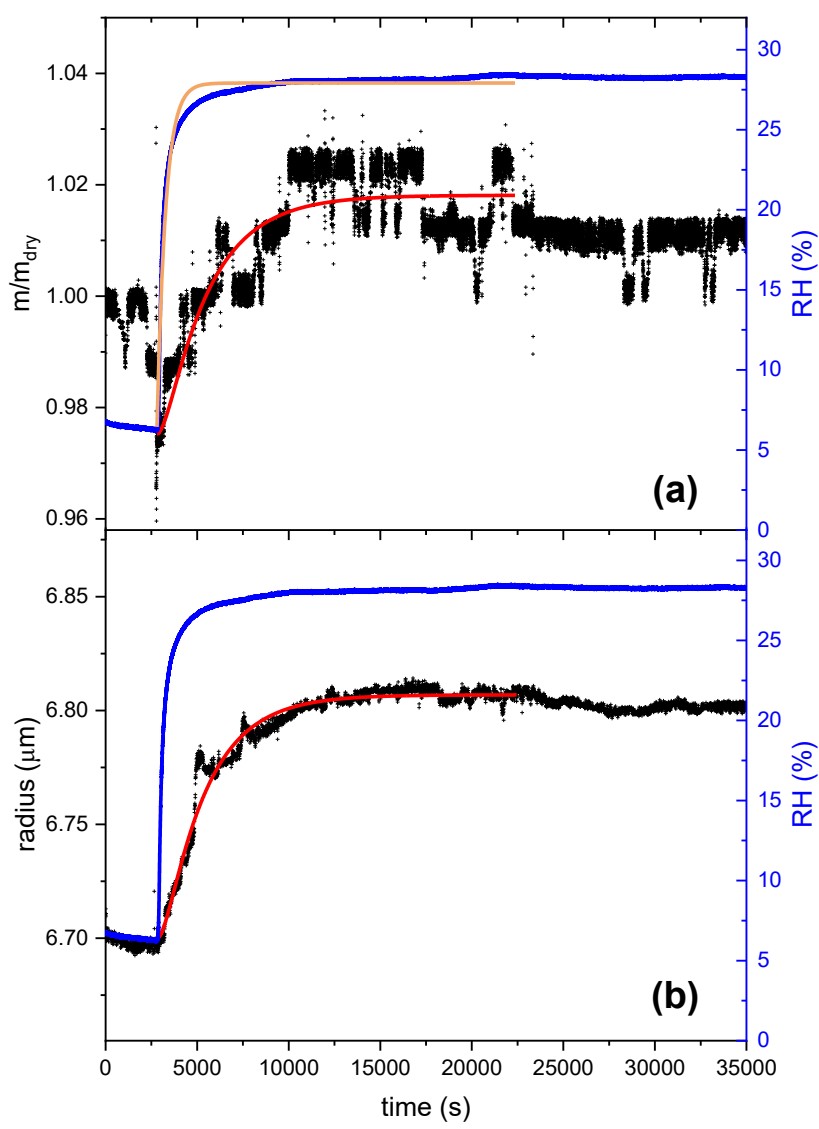





**Figure 6: The response of AA to rapid changes in RH after aging with UV and O₃ until 60% mass loss, panel (a) shows mass growth data and panel (b) size data. The blue line in both panels shows the RH data (right scale). At time t = 2,804 s, the gas flow was switched from dry to humid with RH increasing rapidly from 6.3% to 28.2% and afterwards kept constant for 7.5 h. The black lines represent the response of the aged AA particle in terms of mass (a) and size (b) to the rapid increase in RH. The orange line in panel (a) is an exponential fit to the RH data. The red lines are the first-order kinetics fitting on the mass response (a) and size (b) response. Fitting was done only up to 22,300 s because after this time the mass and size of the particle started to decrease slightly, possibly due to volatilization of some of the remaining products when RH was increased. $\tau_2$, determined from the RH-fit, was found to be 472 s. $\tau_1$, determined from the fitting according to Eq. (4), was found to be 2,503 s for the mass response and 2,312 s for the size response. Note: the step-like noise seen in the mass growth data is an artifact of the instrument, mainly due to the automatic feedback loop that adjusts the voltage to keep the particle at a fixed position, but with a finite resolution in both position and voltage. The dip in mass growth seen just before the RH step is due to sudden changes in the flow system when turning on the humid flow after drying.**

## 3 Results and Discussion

### 3.1 Method validation for the retrieval of diffusivity coefficient of water

To validate our approach of diffusivity coefficient retrieval, $D$ was determined for the same experiment shown in Fig. 6 using the numerical model described in detail in (Zobrist et al., 2011). The model treats the particle as consisting of up to several thousand individual shells and uses a composition and temperature dependent parametrization of the diffusion coefficient of water, simulating the growth and shrinkage of the particle resulting from water diffusion between the shells.

The model was driven by experimental RH(t) data while drying from 28% to 6% and then humidifying up to 28% as well as the initial radius. For the input, several parametrizations for the water activity dependent diffusivity were tested (see Fig. 7). Parametrization for every input $D$ was derived by linear extrapolation of diffusivity in pure water ($2 \times 10^{-5}$ cm² s⁻¹) and input $D$ at $a_w = 0.17$ (ranging from $1 \times 10^{-12}$ cm² s⁻¹ to $1 \times 10^{-11}$ cm² s⁻¹), assuming that the $a_w$ dependence of $D$ is linear on a log scale.

As shown in Fig. 7, considerable agreement between experimental and modelled radius data is obtained when the input $D$ was set between $3 \times 10^{-12}$ cm² s⁻¹ and $8 \times 10^{-12}$ cm² s⁻¹. $D$ derived for the same particle using the linear response approximation (fitting according to Eq. (4)) is $3 \times 10^{-12}$ cm² s⁻¹. In addition, it can be clearly seen that while in the drying part the radius data retrieved from the model for the input $D = 3 \times 10^{-12}$ cm² s⁻¹ does not agree well with the experimental data, it agrees quite well after the step increase in RH. This is consistent with $D$ derived from the fitting ($3 \times 10^{-12}$ cm² s⁻¹), which is done in the same RH range, i.e. the step increase. Thus, $D$ derived using the two different approaches agree within a factor of 2, which is chosen as the uncertainty range in our determination of $D$.

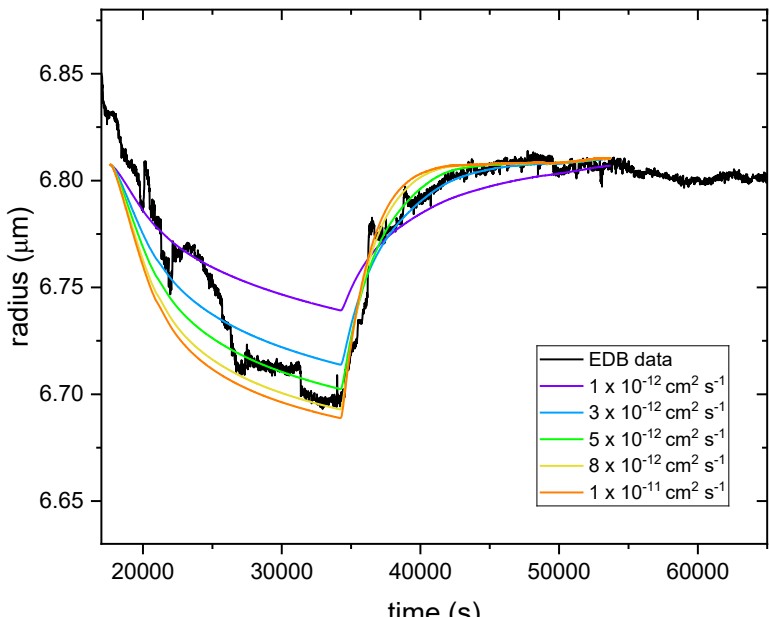

274

**Figure 7: Radius data retrieved from the numerical model for different input D($a_w$) (colored lines) in the range of 1 x
10$^{-12}$ cm$^2$ s$^{-1}$ (violet line) to 1 x 10$^{-11}$ cm$^2$ s$^{-1}$ (orange line) for D($a_w$ = 0.17) in comparison to the experimental radius data
(black line) for AA particle subjected to a rapid increase in RH after aging with UV and O$_3$ until 60% mass loss. Note:
in the modeled data, the initial radius at RH 28% (for all input D) is slightly lower than the equilibrium radius at the
same RH after the step increase. This is because of the uncertainty in size data, as described in Appendix A.**

280

### 3.2 Kinetics: comparison between ozonolysis and photolysis

Figure 8 shows the normalized mass remaining calculated from the voltage data compensating for the gravitational
force as shown in Fig. 4. The mass of an aqueous AA particle remains constant in the absence of UV irradiation at
375 nm or O$_3$ exposure, indicating that AA has a low vapor pressure and does not undergo any reaction in the
presence of oxygen. In the presence of UV irradiation and/or O$_3$ exposure, mass loss is observed, which is evidence
for the occurrence of fragmentation reactions that eventually lead to the formation of volatile products that partition
from the particle phase to the gas phase. Fragmentation reactions are known to be an important pathway for organic
aerosol particles undergoing photochemical reactions as well as for particles exposed to oxidants (Dou et al., 2021;
Kroll et al., 2015; Sun and Smith, 2024; O'brien and Kroll, 2019). However, as shown in Fig. 8, only slight mass
loss is observed when photolysis is done under a nitrogen gas phase, highlighting the importance of oxygen in the
photolysis mechanism. Acceleration in mass loss during photolysis of SOA when transitioning from nitrogen to
zero air was also observed previously in a different study (Sun and Smith, 2024).

The decay kinetics of an AA particle upon reaction with ozone is linear with time, similar to a previous study
(Willis and Wilson, 2022), which indicates that the ozonolysis reaction follows a zeroth-order kinetics. The slow
kinetics observed for ozonolysis may be due to the conjugated structure of AA, which stabilizes the C=C double
bond. According to the study by Wills and Wilson (2022), most of the reactions between O$_3$ and AA occur in the
particle bulk due to the slow reaction of AA with O$_3$ and AA's low surface affinity.

In contrast to ozonolysis, the shape of the photolysis decay kinetics is not linear. It is slow in the beginning, then
speeds up possibly due to the formation of more photoreactive intermediates or due to the initiation of an
autocatalytic process. After about 60% mass loss, the reaction slows down again, most likely because most



photoreactive species are consumed and non-absorbing products are formed or because of radical recombination.
We conclude that similar to the findings by (O'brien and Kroll, 2019) and (Sun and Smith, 2024), initial photolytic
mass loss cannot be extrapolated to the entire SOA mass loss and that a photo-recalcitrant fraction remains, which
prevents or slows down further mass loss.
Moreover, it can be inferred from Fig. 8 that the rate of decay of AA is slightly faster when photolysis is done in
the presence of $O_3$. Since ozone is a stronger oxidant compared to oxygen, it might result in a faster mass loss,
especially in the beginning of the reaction, where the oxidant is most likely the limiting factor. After the initial
induction period, the rates of both reactions become comparable because of the occurrence of acceleration in the
absence of $O_3$.
Additionally, regarding the photolysis experiments, it can be inferred from Fig. 9 that the rates of reaction of two
different AA particles with UV irradiation overlap in the beginning of the experiment, but acceleration in mass
loss of particle 1 occurs before particle 2. For instance, after an exposure time of 40,000 s, particle 1 lost 64% of
its mass whereas particle 2 lost 48% (see Fig. 9). This is most likely because oxygen is the rate limiting factor in
the beginning of the reaction. As the reaction proceeds, the intensity of the UV irradiation becomes important,
which results in different reaction rates. This might be due to slightly different particle sizes and the position of
the particle relative to the laser beam, which can affect the photon flux into the particle and thus the reaction rate.
For this reason, mass loss was used instead of exposure time to represent the extent of aging, as will be presented
in the following sections. However, mass loss and exposure time are well correlated (see Fig. E-1 in the appendix),
but mass loss was chosen to account for the differences between different particles in terms of the reaction rate,
assuming that at a certain  mass loss, rather than exposure time, different particles should have similar chemical
composition despite having different reaction rates.

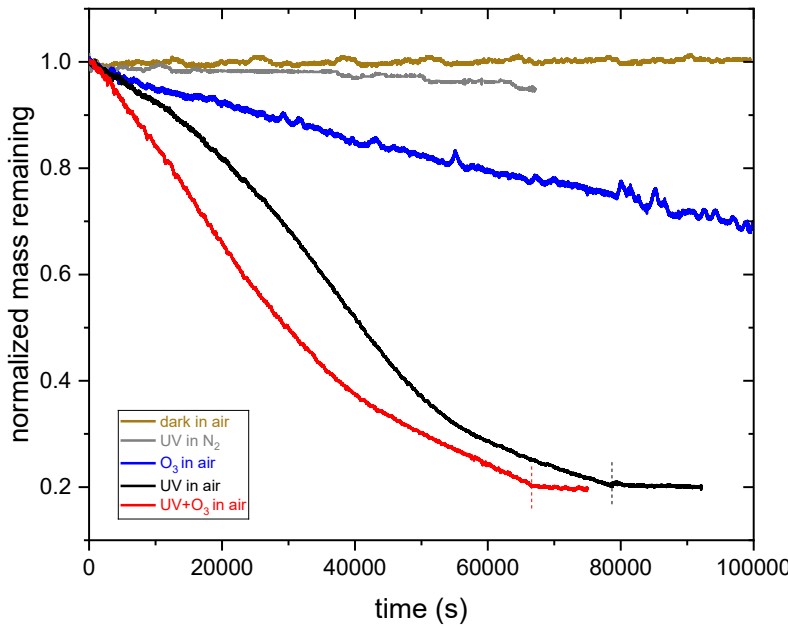


**Figure 8: Comparison of the decay kinetics of an AA particle exposed to 375 nm UV irradiation in pure nitrogen and in synthetic air, 10 ppm $O_3$ in synthetic air, and UV+$O_3$ in synthetic air. Vertical dashed lines correspond to switching off UV (black) and UV+$O_3$ (red).**




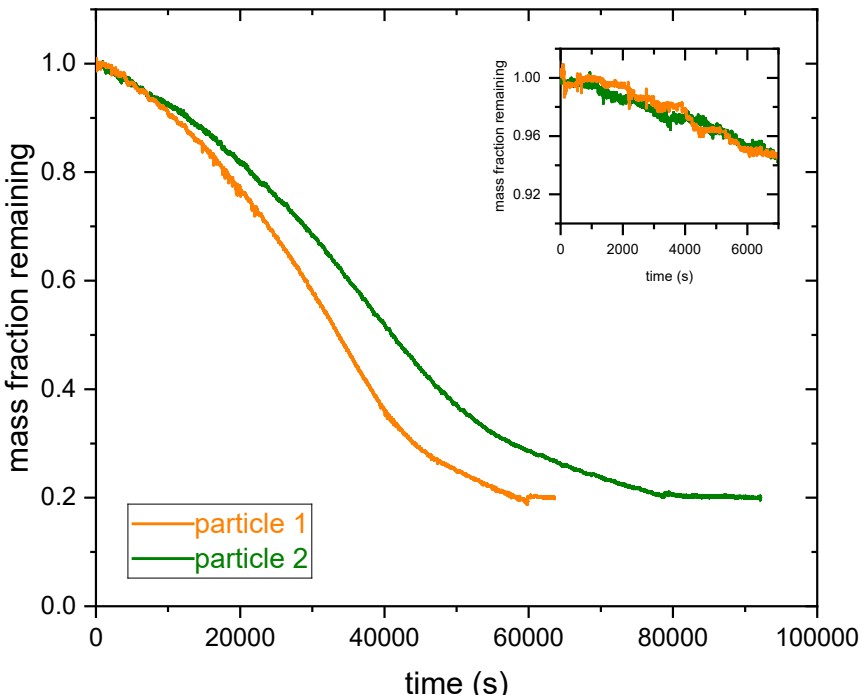


**Figure 9: Comparison of the decay kinetics between two different AA particles exposed to 375 nm UV irradiation until 80% mass loss. The inlet graph shows the first 7,000 s to emphasize the overlap in the initial reaction rate between the two experiments.**


### 3.3 Phase separation during ozonolysis

Figure 10 shows the mean fringe distance (a) and pattern distortion parameter (b) of a 532 nm laser-illuminated AA particle upon exposure to ozone. After about an hour of $O_3$ exposure, the TAOS pattern of the particle becomes irregular and the pattern distortion parameter, which denotes the asymmetry in the scattering pattern (Braun and Krieger, 2001), becomes larger. This indicates that the particle is no longer homogenous and spherical and has lost its symmetry, resulting in light scattering with different intensities depending on its orientation relative to the laser beam. This behavior was observed only upon exposure to $O_3$ and was confirmed through five other $O_3$ exposure experiments, where an irregularity in the TAOS pattern was observed for all particles after an exposure time of 65 ± 8 min. On the contrary, AA particles exposed to UV illumination or UV and $O_3$ simultaneously remained spherical symmetric upon aging. The observed heterogeneity in the particle upon ozonolysis could be due to liquid-liquid phase separation (LLPS) exhibiting partial wetting morphology. Similar behavior was observed in other studies upon ozonolysis of oleic acid particles (Hosny et al., 2016) and squalene droplets (Athanasiadis et al., 2016). However, in the case of ozonolysis of oleic acid, they observed a wide range of species: small oxidized polar molecules and large non-polar oligomers, which are immiscible and can favor LLPS. Here, we did not perform chemical characterization of the ozonolysis products, but (Willis and Wilson, 2022) identified products having two to seven carbon atoms with different carbon oxidation states (~1 to 3) in AA ozonolysis experiments. It might be possible that some of the small products with high carbon oxidation state are immiscible with the larger products with low oxidation state, which can cause phase separation. Another possibility for the observed symmetry loss in light scattering would be precipitation of *trans*-aconitic acid or one of the ozonolysis products that is insoluble in water, creating two phases: solid precipitate and dissolved liquid phase, since the particles contain significant amount of water as they are kept at relatively high RH (above 78%).



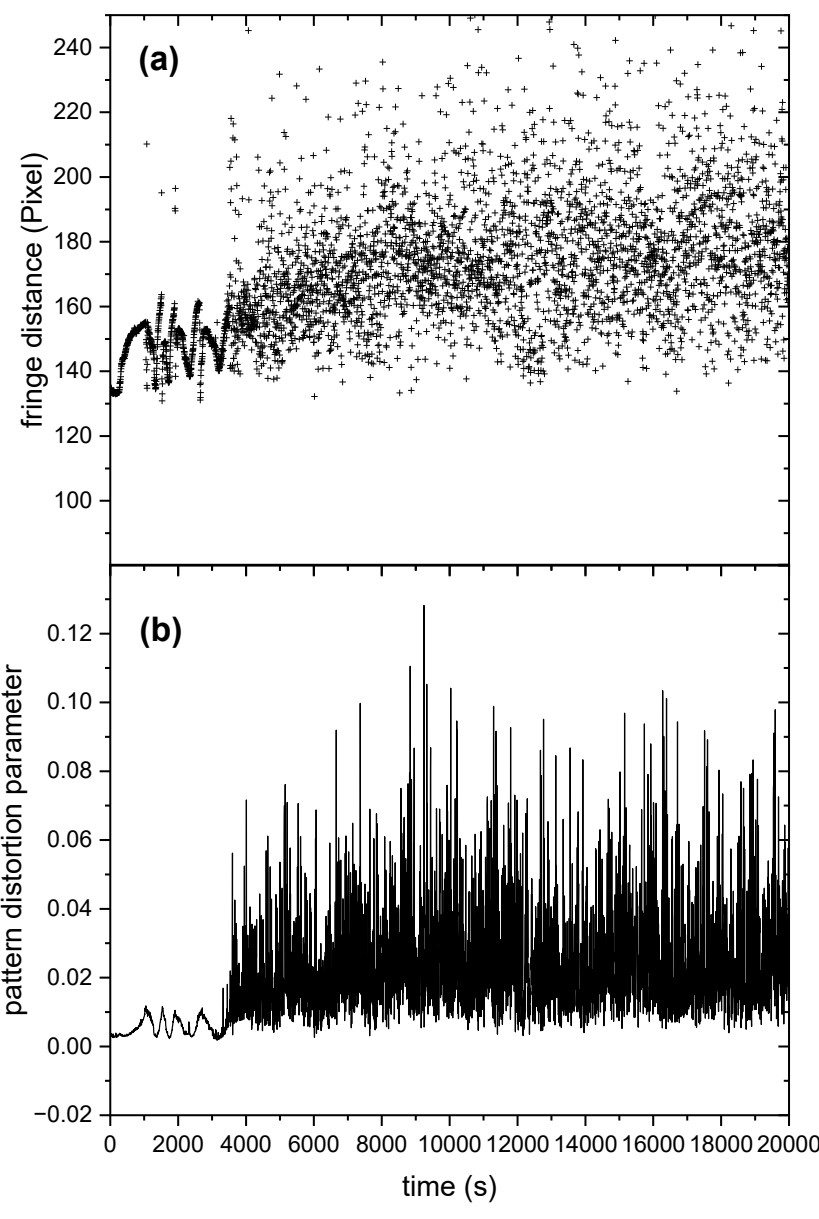



**Figure 10: (a) Mean fringe distance and (b) pattern distortion parameter of 532 nm laser-illuminated AA exposed to O$_3$.**
**Only the data for the first 20,000 s are plotted here to present clearly the phase transition occurring at ~ 3,500 s, indicated**
**by the significant increase in fluctuations in fringe distance (a) and pattern distortion parameter (b).**

**3.4 Phase state of AA after aging**
In contrast to untreated AA, which effloresced typically at RH 59.9%, AA particles aged with UV or UV and O$_3$
simultaneously did not crystallize upon drying. This was deduced from the TAOS pattern, which remained regular
upon drying, implying that the aged particles remained liquid or amorphous solid, but not crystalline. After AA
particles undergo chemical reaction, a mixture of products is produced, which makes efflorescence less likely to
occur due to thermodynamic reasons.
Similar behavior in terms of suppression or inhibition of efflorescence after aging was observed previously. For
instance, suppression of crystallization was observed upon ozonolysis of mixed maleic acid/ammonium sulfate
particles and subsequent drying (Chan and Chan, 2012). In another study, maleic acid particles exposed to O$_3$
showed efflorescence at lower RH compared to unprocessed maleic acid particles, with some particles not
effloresing at any RH (Pope et al., 2010).
Furthermore, for AA particles aged with UV or UV and O$_3$ only up to 20% or 26% mass loss, although the particles
remained spherical upon drying, the TAOS pattern became irregular after RH was increased rapidly from 6% to
28%, indicating crystallization of AA (see Appendix F). This could be due to restructuring the hydrogen bonding
network favoring nucleation. Alternatively, if nucleation already occurred during drying, the plasticizing effect of
water could lead to crystal growth after water uptake. This behavior was not observed for particles that lost more
than 25% of their mass because of the conversion of considerable amount of *trans*-aconitic acid into a mixture of
products, which inhibited crystallization.

**3.5 Viscosity and hygroscopicity of AA before aging**
The average viscosity of untreated AA derived from our experiments at room temperature (~298K) and RH 17.7
$\pm$ 0.5% (midpoint of the step in RH 6–28%) is 1.2 x 10$^3$ Pa s. Aconitic acid and citric acid are structurally similar:
both are tricarboxylic acids having 6 carbon atoms. (Song et al., 2016) measured the viscosity of citric acid by
droplet coalescence at ambient temperature and reported as 5.1 x 10$^3$ Pa s at 18 % RH, which is within a factor of
4 of the viscosity of AA determined in this study. The slightly higher viscosity of citric acid compared to aconitic
acid might be due to the additional hydroxyl group, which enhances hydrogen bonding (Rothfuss and Petters,
2017) and the lower temperature of the coalescence experiment.
Figure 11 shows the hygroscopicity results in terms of derived $\kappa$ values. For reference, a selection of $\kappa$ values of
other organic acids are taken from the literature. As shown in this figure, organic acids can have quite different $\kappa$
values. Higher $\kappa$ values indicate higher hygroscopicity. Hygroscopicity of organic compounds depends on
molecular structure and physicochemical properties. Organic compounds having more functional groups, such as
carboxyl, carbonyl, and hydroxyl tend to be more hygroscopic (Han et al., 2022). In general, polar organic
compounds having higher O:C are more hygroscopic (Han et al., 2022; Lambe et al., 2011; Duplissy et al., 2011).
In addition, organic compounds that are more water soluble tend to be more hygroscopic because they would have
a higher molar concentration in the saturated solution corresponding to a stronger reduction in water activity (Han
et al., 2022).
Average $\kappa$ of untreated AA measured in this study is 0.254 $\pm$ 0.031, which agrees within error with *cis*-aconitic
acid (0.21) determined in another study (Han et al., 2022) and that predicted by AIOMFAC model (0.237) (Zuend
et al., 2011; Zuend et al., 2008), but is significantly higher than that of *trans*-aconitic acid (0.172 $\pm$ 0.010)
determined in another study (Rickards et al., 2013). In our study, fitting was done between RH 62% and 80%,
which is different from the RH used in Rickards et. al (> 90%), which could be one reason for the difference.
Variations in $\kappa$ for the same organic acid between different studies have been observed previously. For instance,
as shown in Fig. 11, $\kappa$ for citric acid determined using a humidified tandem differential mobility analyzer
(HTDMA) at RH 90% was found to be 0.18 (Han et al., 2022), while that determined using aerosol optical tweezers
(AOT) at RH 66% was found to be 0.233 $\pm$ 0.035 (Rickards et al., 2013). In the following, we focus on changes
in $\kappa$ with aging rather than on absolute $\kappa$ values.




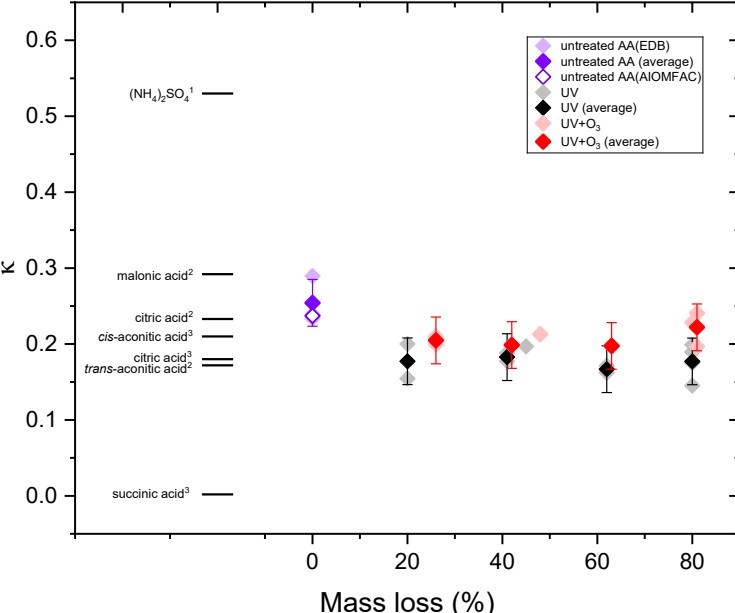



**Figure 11: The hygroscopicity of AA particles before and after UV and UV+O$_3$ aging in comparison with (NH$_4$)$_2$SO$_4$, malonic acid, citric acid, cis-aconitic acid, trans-aconitic acid, and succinic acid data taken from the literature: (1) (Petters and Kreidenweis, 2007) (2) (Rickards et al., 2013) (3) (Han et al., 2022).**

### 3.6 Viscosity and hygroscopicity of AA after aging

Figure 12 shows the viscosity of untreated AA and AA aged with UV and UV and O$_3$ simultaneously up to 20%, 40%, 50%, 60%, and 80% mass loss. All measurements were done at room temperature (~298 K) and at RH 17.7 ± 0.5%. The viscosity of AA exposed to UV and O$_3$ simultaneously increased with the extent of aging, with the aged AA at 80% mass loss being almost 4 orders of magnitude higher compared to that of the untreated AA. UV exposure alone also resulted in an increase in viscosity but only up to 60% mass loss. Surprisingly, further UV-aging (80% mass loss) led to a reduction in viscosity. However, with UV-aging at 80% mass loss, there is a large spread in the viscosity data, where some of the values at the lower viscosity range were beyond the limit of determination with our method (see Appendix D). Since the discrepancies in the viscosity values are mainly between size and mass measurement for the same particle, rather than being between different experiments, the uncertainty is most likely due to experimental error at 80% mass loss, where the particle becomes small, increasing the error on both size and mass measurements. Thus, it is difficult to conclude the exact viscosity at 80% mass loss, but it is evident that the viscosity in all cases is significantly less than the one at 60% mass loss.

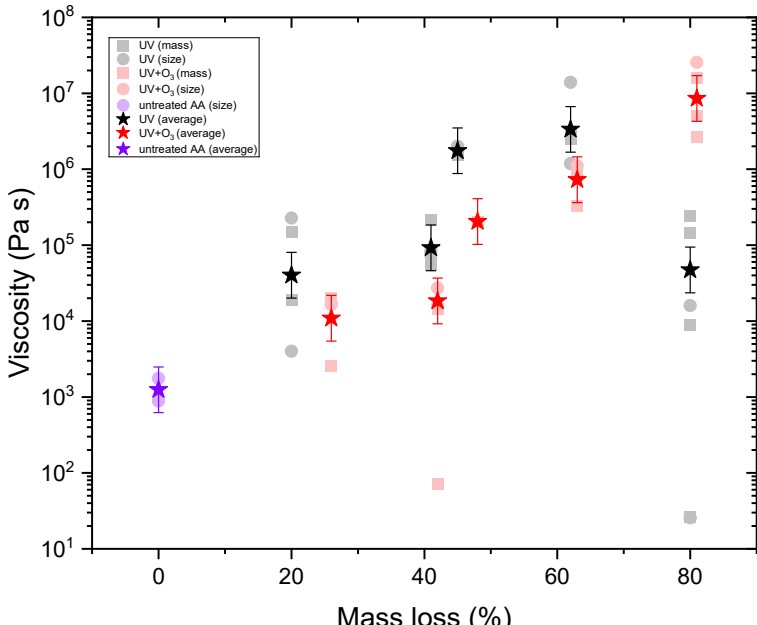


**Figure 12: The viscosity of AA particles before and after UV and UV+O₃ aging from mass and/or size data.**


As shown in Fig. 11, average κ of aqueous AA particles decreased slightly from $0.254 \pm 0.031$ to $0.177 \pm 0.031$
upon aging with UV light till 20% mass loss. Upon further aging, κ remained almost constant. For the AA particles
aged with UV and O₃ simultaneously, less significant reduction in κ was observed. Similar to UV-aging, κ
decreased from $0.254 \pm 0.031$ to $0.205 \pm 0.031$ after 25% mass loss, but then it remained almost constant with
further aging. However, this reduction is not significant, taking the uncertainty in κ into account. In addition,
overall κ values for UV+O₃ aged AA are a bit higher than those of UV-aged AA, consistent with the viscosity
values, which show slightly lower viscosity for UV+O₃ aged AA up to 60% mass loss.
Currently, we do not have any information about the chemical composition of the particles after aging. However,
we may draw some conclusions from the data shown in Fig. 11 and 12. Since hygroscopicity did not increase after
aging, O:C is also expected to remain almost constant or decrease slightly after aging. While it is known that
hygroscopicity and carbon oxidation state increase after oxidative aging (Jimenez et al., 2009; Kroll et al., 2011;
Kroll et al., 2015), no significant or small net change in O:C was observed after photolysis of SOA in other studies
(Romonosky et al., 2015; O'brien and Kroll, 2019). It should be noted that in this study, only the hygroscopicity
and viscosity of the condensed phase after aging are measured, without considering the highly volatile compounds
evaporating to the gas phase. It is still expected that the overall carbon oxidation state increased after aging in our
study if the compounds lost to the gas phase are also considered. As gas phase oxidants are involved in the aging
process (O₂ in UV-aging and O₃ in UV+O₃ aging), it is expected that polar oxygen containing functional groups
are added after oxidation. Since these groups weaken adjacent C-C bonds, fragmentation reactions will take place,
leading to volatilization of smaller fragments to the gas phase. This mechanism can be indirectly inferred from
Fig. 8, where initially an induction period appears, followed by rapid mass loss. However, it is expected that the
most highly oxidized carbon will exist mainly in the gas phase (Kroll et al., 2011). Therefore, although the small,
fragmented products are expected to have higher oxidation state, they are also expected to be volatile, thus the
oxidation state of the particle can decrease slightly after these fragments evaporate (O'brien and Kroll, 2019;
Hildebrandt Ruiz et al., 2015). However, we cannot prove that O:C remained constant or decreased slightly, since



we did not determine the chemical composition of the products. Note that if O:C increased after aging, but if the
average molecular weight increased as well, the hygroscopicity may still decrease.
Viscosity increase upon SOA aging was observed previously in the UV-aging of d-limonene and α-pinene SOA
(Baboomian et al., 2022). Mass spectrometric analysis indicated that the increase in d-limonene SOA viscosity
was most likely due to changes in the chemical properties because an average increase in molecular weight,
elemental O:C ratio, number of carbon atoms per molecule, and double bond equivalent was observed (Baboomian
et al., 2022). However, in contrast to our study, photolysis of SOA was done at low RH. Water content of the
particle during photolysis can play an important role in the mechanism of formation of products and thus in the
viscosity of the final products. For instance, when photolysis of SOA was performed in aqueous solution instead
of low RH, reduction in carbon number was observed, which was attributed to the degradation of dimers and
trimers (Romonosky et al., 2015). Another study comparing condensed-phase SOA photolysis at low RH and
aqueous photolysis found that the signal intensity obtained by mass spectrometry in the larger m/z range remained
similar before and after exposure, while lower m/z range was higher after aqueous photolysis, but lower after
condensed-phase photolysis (Sun and Smith, 2024). They attributed this difference to the evaporation of the
volatile products from the condensed phase, but retention of these species in the aqueous solution. Direct
comparison of our study to these studies is difficult, since in our case condensed-phase photolysis was done at
higher RH. However, since mass loss was observed and the viscosity increased after UV-aging, it can be inferred
that the smaller volatile fragments evaporated, leaving relatively larger non-volatile products in the particle.
One possible explanation for the observed viscosity enhancement after aging is the formation of oligomers from
product fragments. Oligomers, due to their high molecular weight, could contribute to increased viscosity.
However, the formation of oligomers might be hindered under the experimental conditions, as the reactions occur
at moderately high RH, and the presence of water typically slows down oligomerization. It is important to note,
though, that the particles are more concentrated in water at these RH levels (even at 85%) compared to typical
aqueous-phase chemistry, which may facilitate oligomer formation. Additionally, oligomerization could have
occurred during the drying process just before viscosity measurements were taken.
Alternatively, the increase in viscosity could be attributed to changes in the hydrogen bonding network. The
intermolecular hydrogen bonding between the products might be enhanced compared to untreated AA, leading to
a higher viscosity. Further UV-aging could degrade some of these products or the oligomers, potentially altering
the hydrogen bonding interactions and causing a decrease in viscosity upon continued aging.
The continuous increase in viscosity upon aging with UV and $O_3$ simultaneously until 80% mass loss suggests that
the aging mechanism and the products formed are different in the presence of oxidants like $O_3$. The presence of
$O_3$ might favor the formation of oligomers, therefore increasing the viscosity of the particles further with longer
exposure.

## 4 Conclusion

In this study, the viscosity and hygroscopicity of aqueous *trans* aconitic acid particles were determined after
exposure to 375 nm UV irradiation or UV irradiation and $O_3$. Photolysis and ozonolysis resulted in significant
mass loss because of fragmentation reactions followed by volatilization and displayed different reaction kinetics.
Ozonolysis alone disrupted the light scattering symmetry of the particle, indicating a phase transition. Aging with
UV or UV and $O_3$ resulted in the inhibition of efflorescence upon drying. The viscosity of the particles increased
by almost 4 orders of magnitude upon aging with UV and $O_3$ simultaneously. UV-aging also resulted in an increase
in viscosity but only up to 60% mass loss. Further UV-aging resulted in a reduction in viscosity. Hygroscopicity
of the particles decreased slightly after exposure to UV laser up to 20% mass loss and then stayed constant with
further aging. A similar trend, though less significant, was observed for combined UV and $O_3$ aging.
To check the relevance of our study to atmospheric conditions, we estimated the characteristic mixing times within
the particles after aging. The increase in viscosity upon aging implies for particles with a radius of 100 nm (typical
of accumulation mode SOA particles) a characteristic mixing time of 4 h compared to 4 s for the unexposed
particles under dry boundary layer conditions. For colder temperatures in the free troposphere, the mixing times
could be still longer even at higher relative humidity.
Therefore, the changes in viscosity, hygroscopicity, phase state, volatility, and mixing times observed in this study
provide further evidence on how aging can directly alter the physicochemical properties of aerosol particles, with



indirect consequences on climate and human health. Our results show that the presence of gas phase oxidants like
ozone, in addition to UV light, results in viscosity enhancement, an effect seen previously in the UV-aging of SOA.
This viscosity enhancement is most likely due to the formation of oligomers of high molecular weight. However,
from the shape of reaction kinetics, hygroscopicity, and viscosity results, we conclude that the mechanism of aging
is different in the presence of gas phase oxidants compared to photochemical aging without gas phase oxidants.
To understand better the mechanism behind the differences that we observe, we plan to study in the future the
chemical composition of the products during aging using mass spectrometry.
































**Appendix A.** Size retrieval from two-dimensional angular optical scattering (TAOS) pattern
The EDB configuration and geometrical constraints allow us to measure the TAOS pattern within about 5° half
apex angle around 100° scattering. Figure A-1 shows four TAOS-patterns observed in our experiments.

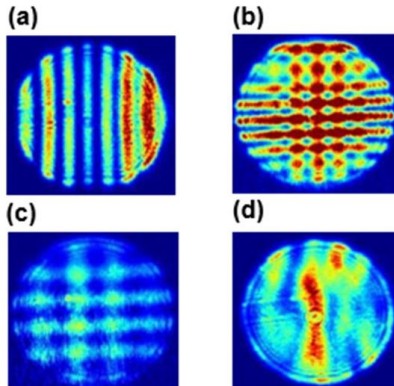


**Figure A-1: (a) TAOS pattern of a spherical symmetric trans-aconitic acid particle upon illumination with polarized**
**532 nm laser, fringe intensity is rainbow color coded in arbitrary units. (b) pattern of a particle upon illumination by**
**both the polarized 532 nm laser and the orthogonally polarized 375 nm laser. (c) same as (b) but after photochemical**
**aging. (d) TAOS pattern for 532 nm illumination only after phase transition induced by ozonolysis.**
Comparing Fig. A-1(d) with the other panels of Fig. A-1 illustrate that the regular TAOS pattern is distorted once
the spherical symmetry of a particle is lost. In comparison to Fig. A-1(b), the average fringe distance for both
orthogonal fringe patterns in Fig. A-1(c) has increased, indicating shrinkage during aging. Quantitative analysis of
these types of patterns, i.e. size retrieval, has been discussed extensively in the literature (Davis and Periasamy,
1985; Steiner et al., 1999; Jakubczyk et al., 2013; Davies, 2019). A fast, simple method for sizing a spherical
symmetric particle is based on scattered light angular frequency determination (Steiner et al., 1999), which requires
a calibration of the TAOS pattern in terms of scattering angle. We follow the suggestion by (Davies, 2019) and
measure the evaporation rate of a pure compound for calibration, here PEG-4 for which the vapor pressure is well
established (Krieger et al., 2018).
Sizing by measuring the characteristic angular frequency works best for scattering angles smaller than 60°.
However, our scattering angle is 100° and our apex angle is quite small due to spatial restriction in our setup. This
limits its applicability to small size changes, as the uncertainty in sizing using the characteristic angular frequency
is of the order of ± 1 μm for our scattering geometry and typical particle size. Therefore, for measuring small size
changes, we apply a method based on spectral shift of Mie-resonance spectra (Zardini et al., 2006) to angular shift
of a peak position in the TAOS pattern. Compare this approach with a related method based on tabulated peak
positions suggested by (Davies, 2019). Figure A-2 shows data taken for an evaporating PEG-4 particle at constant
temperature. Panel (a) shows that the radius scales almost linearly over a wide size range with the total peak
position shift in the TAOS pattern. On closer inspection (see panel (b) of Fig. A-2), the relationship is irregular on
small scales but allows the reliable detection of size changes of about ± 40 nm. In our application, we are focusing
on characteristic time scales of small size changes due to rapid changes in relative humidity: there, the irregularity
between size and peak position shift for small size changes influences the determination of characteristic time only
slightly (see Fig. 6 in main text.)

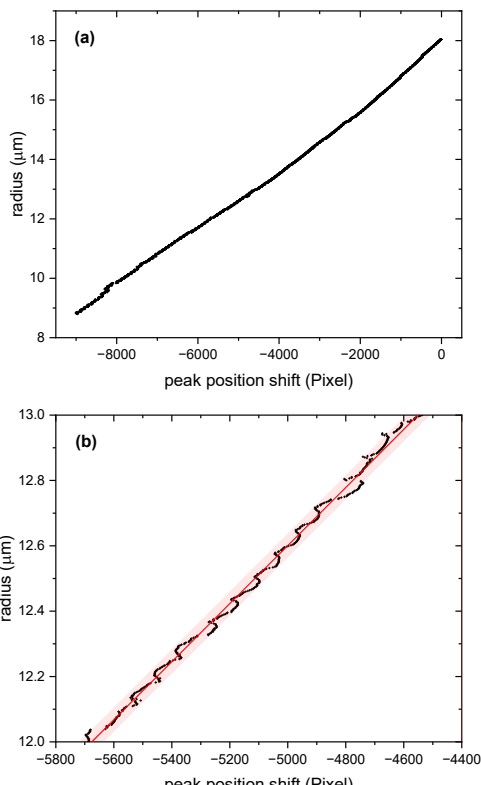


**Figure A-2: (a) Size data of an evaporation experiment (PEG-4 at 24°C) versus peak position shift. Evaporation rate**
**was calculated from known data of vapor pressure (Krieger et al., 2018) and radius was determined through comparison**
**of measured characteristic angular frequency with evaporation rate. (b) Zoom into the data of (a) showing that radius**
**can be measured with a sensitivity of about ± 40 nm when measuring the peak position shift. Note however, that the**
**relationship between peak position shift and size is not strictly linear, but irregular within a size change of approximately**
**40 nm.**

575

**Appendix B.** Density determination of aqueous AA

To determine the density of aqueous AA under dry conditions ($\rho_0$), the density of AA at mass fraction $w_s = 0.286$
was measured using a pycnometer. Assuming the conventional additivity rule holds (Lienhard et al., 2012; Steimer
et al., 2015):

$$\rho(w_s) = \left( \frac{1-w_s}{\rho_w} + \frac{w_s}{\rho_0} \right) \tag{B1}$$

where $\rho(w_s)$ is the density of solute at mass fraction $w_s$, $\rho_w$ is the density of water, we retrieve $\rho_0$, the density of
aqueous AA solute at dry conditions, i.e. the subcooled melt density. Based on the known density of water and our
measured density at $w_s = 0.286$, $\rho_0 = 1.56$ g cm$^{-3}$.


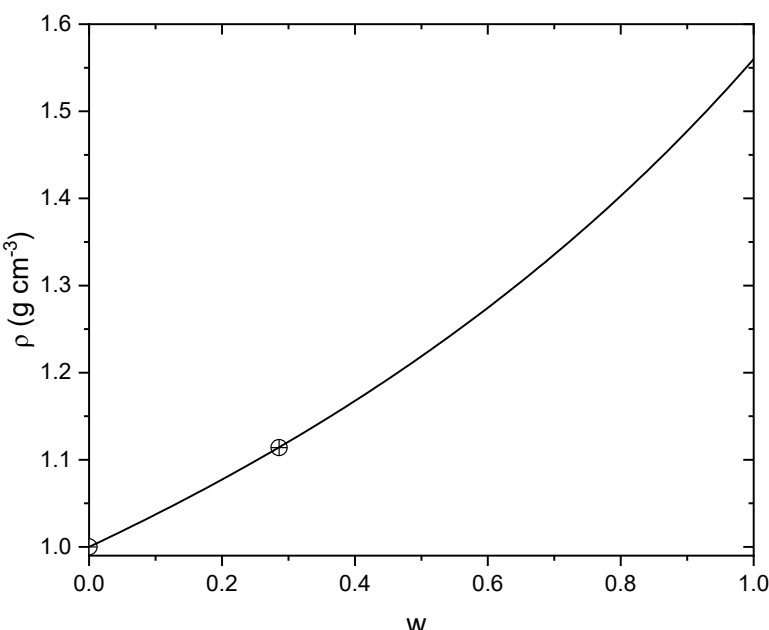


**Figure B-1: Density ($\rho$) vs. mass fraction ($w_s$) parametrization for aqueous AA according to Eq. (B1).**


**Appendix C.** Deconvolution of linear response to an exponential 'step' in relative humidity, derivation of Eq. (4)
Let's assume the dynamic system is a linear time-invariant one, with the response of the system to a step input
(Heaviside function, $H(t)$) is given by:
$$y(t) = 1 - e^{-k_1 t}$$
Now, instead of a step input, $H(t)$, the input function is given by:
$$RH(t) = 1 - e^{-k_2 t}$$
In our experiment, we obtain $k_2$ by analyzing the data $RH(t)$, with $k_2$ being the rate constant of the RH-step. We
want to determine the system's response $y(t)$ to the input $RH(t)$. Very generally, the system's response is given
by the convolution integral:
$$y(t) = \int_0^t RH(\tau) h(t - \tau) \, d\tau$$
where the impulse response $h(t)$ is the derivative of the system to a step response $H(t)$:
$$h(t) = k_1 e^{-k_1 t}, \quad t > 0$$
For the step response, $H(t)$, the radius or mass evolution is related to the impulse response by:
$$y(t) = \int_0^t h(\tau) \, d\tau$$
Therefore, substituting the RH input function and the impulse response function by





$h(t - \tau) = k_1 e^{-k_1(t-\tau)}$, we get:
$$y(t) = \int_0^t (1 - e^{-k_2(\tau)}) k_1 e^{-k_1(t-\tau)} \, d\tau$$

Expanding the integral:
$$y(t) = k_1 e^{-k_1 t} \int_0^t (1 - e^{-k_2\tau}) e^{k_1\tau} \, d\tau$$

Solving the integral leads to the final expression for $y(t)$:
$$y(t) = (1 - e^{-k_1 t}) - \frac{k_1}{k_1 - k_2} (e^{-k_2 t} - e^{-k_1 t})$$

Replacing the rate constants with their inverse, i.e. the characteristic time constants $\tau_1 = {}^1/_{k_1}$ and $\tau_2 = {}^1/_{k_2}$,
yields Eq. (4):
$$y(t) = \left(1 - e^{-t/\tau_1}\right) - \frac{\tau_2}{\tau_2 - \tau_1} \left(e^{-t/\tau_2} - e^{-t/\tau_1}\right)$$


**Appendix D.** Comparing the linear response approximation of Appendix C to a numerical model solving the
diffusion equation
As noted in the main text, the linear response approximation for water uptake kinetics is expected to diverge from
the full solution of Fick's second law when water diffusivity becomes sufficiently slow. Here, we compare the size
change resulting from water uptake under an ideal exponential 'step'-like change in RH, using a numerical model
based on the Euler forward method to solve the diffusion equation (Zobrist et al., 2011), with the linear
approximation described in Appendix C.
Since we use the same flow changes in all our experiments and the particle sizes in the experiments vary only over
a rather small size range, we can restrict the comparison to an idealized $RH(t)$ pathway taken from our
experimental observations as well as choosing a particular initial radius. Figure D-1 shows the $RH(t)$ pathway
chosen as a blue line, the size response predicted by the numerical model as a black line and a linear regression of
Eq. (4) to the size predicted by the numerical model as a red line. For simplicity, we assume as input for the
numerical model that the dependence of water diffusivity on water activity follows an exponential dependence for
water activities with the water diffusivity for pure water fixed to 2 x $10^{-5}$ cm$^2$ s$^{-1}$. At 40% RH, an initial radius of
6.969 µm is set and it is assumed that at this starting RH before drying the particle is internally mixed in water
content. For the simulation of Fig. D-1, we set the water diffusivity $D$ ($a_w = 0.17$) to 2 x $10^{-10}$ cm$^2$ s$^{-1}$. As evident
from Fig. D-1 for these parameters, the linear approximation seems to mirror the predictions of the numerical
model very well.
Following this approach, we test the linear response for the range of $D$ ($a_w = 0.17$) between 5 x $10^{-13}$ cm$^2$ s$^{-1}$ to 2 x
$10^{-7}$ cm$^2$ s$^{-1}$. Some of the results are shown in Fig. D-2. It becomes clear that our experiments are not sensitive to
$D$ (at $a_w = 0.17$) > 2 x $10^{-9}$ cm$^2$ s$^{-1}$, simply because our experimental setup yields a rather slow response in $RH(t)$
upon a step change in gas mass flows (We cannot increase the total flow much beyond 50 sccm to allow mass
measurements). For this reason, the second step increase in RH (from 28% to 49%), shown in Fig. 4 in the main
text, was not evaluated. Figure D-2 shows in addition the expected deviation from linear response with low water
diffusivities (panels e and f).
However, we can correct for the differences between numerical model and linear approximation for a limited range
of water diffusivities as shown in Fig. D-3. There, we compare the characteristic time constant, $\tau_1$, obtained from
the linear regression to Eq. (4) to the input parameters of the numerical model represented as characteristic time
according to Eq. (5). Perfect agreement between both is indicated as the dashed line. The limited range of
characteristic response times for which the linear approximation works is clearly visible from Fig. D-3. It allows
us to deduce an empirical correction factor which we apply for characteristic response times measured in our
experiments ranging from 20 s to about 27,000 s. Fortunately, this is approximately the time range we do observe
in our experiments for untreated AA and the most viscous aged particles.




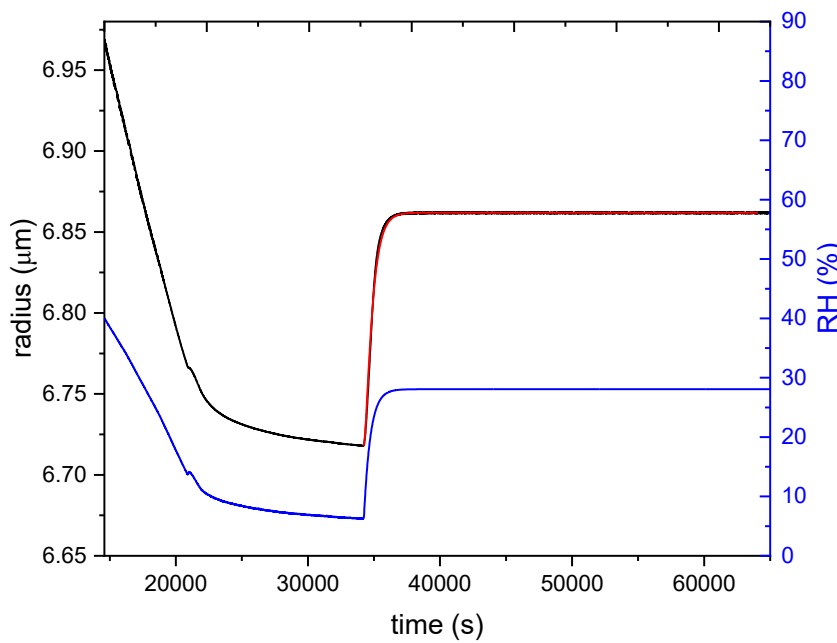


**Figure D-1: The blue line represents the RH data. The black line represents the radius data obtained from the numerical model for an input D of 2 x 10⁻¹⁰ cm² s⁻¹. The red line is the first-order kinetics fitting according to Eq. (4) on the size response data after increasing RH rapidly.**




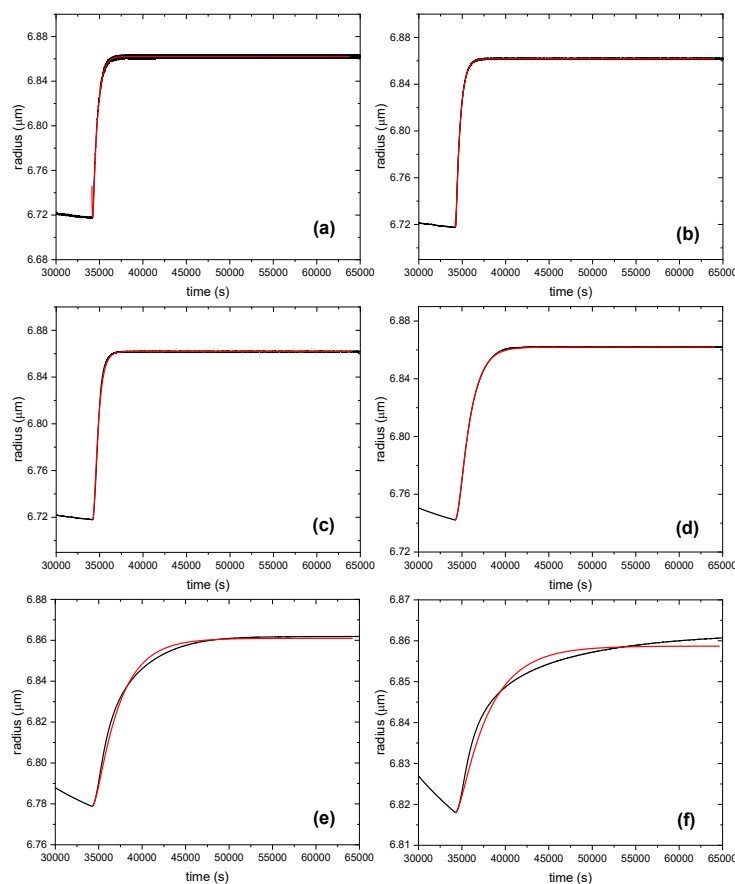


**Figure D-2: The black line represents the size data obtained from the numerical model for an input D (at $a_w$ = 0.17) of (a) 2 x 10$^{-8}$ cm$^2$ s$^{-1}$, (b) 2 x 10$^{-9}$ cm$^2$ s$^{-1}$, (c) 2 x 10$^{-10}$ cm$^2$ s$^{-1}$, (d) 9 x 10$^{-12}$ cm$^2$ s$^{-1}$, (e) 2 x 10$^{-12}$ cm$^2$ s$^{-1}$, and (f) 5 x 10$^{-13}$ cm$^2$ s$^{-1}$. The red line shows the linear response approximation regression curve.**





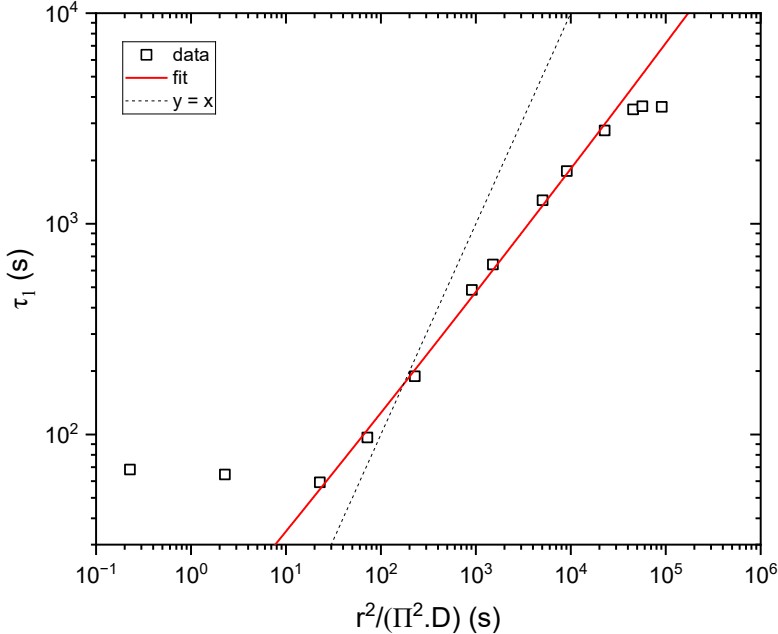


**Figure D-3: Output $\tau_1$ data determined using the exponential function vs. input $r^2/(\Pi^2.D)$ data determined using the numerical model. The red line represents the fit on the output vs. input data. The dashed line shows $\tau_1 = \frac{r^2}{\Pi^2.D}$ .**


**Appendix E.** Correlation between mass loss and exposure time

Figure E-1 shows that as mass loss increases, exposure time also increases for both UV and UV+$O_3$ exposure. This means that mass loss and exposure time are well correlated. Thus, % mass loss can be used to represent the extent of aging.





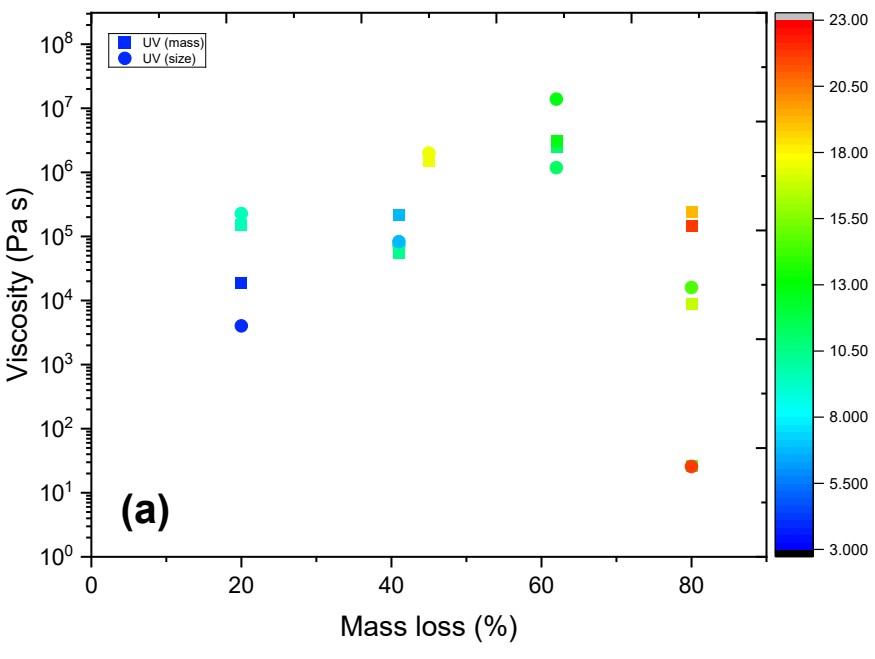

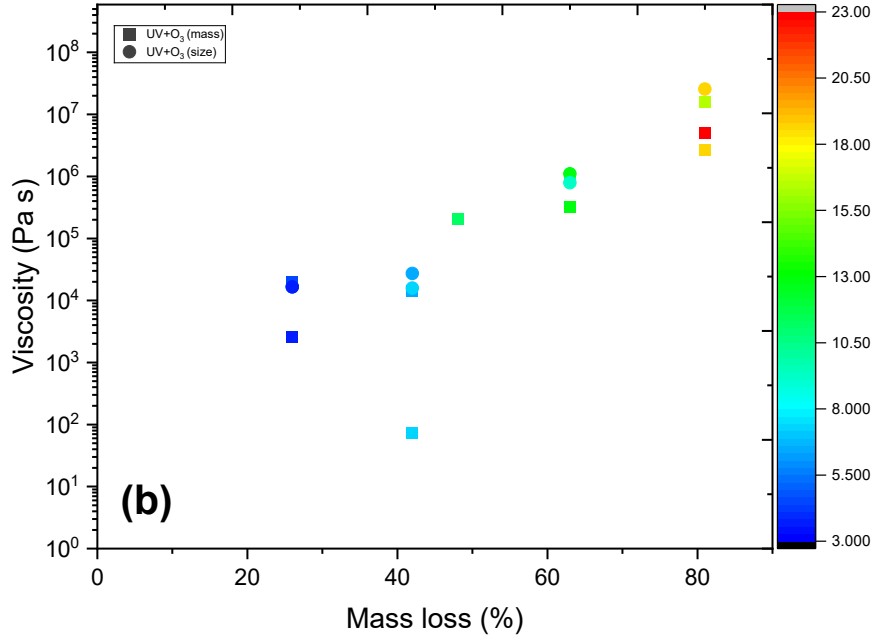



**Figure E-1: Viscosity of AA particles as a function of mass loss color-coded according to the exposure time (h) for (a)**
**UV-exposure experiments and (b) UV+O₃.**

**Appendix F.** Mean fringe distance and pattern distortion parameter of AA particle exposed to UV and $O_3$ up to
26% mass loss upon humidifying



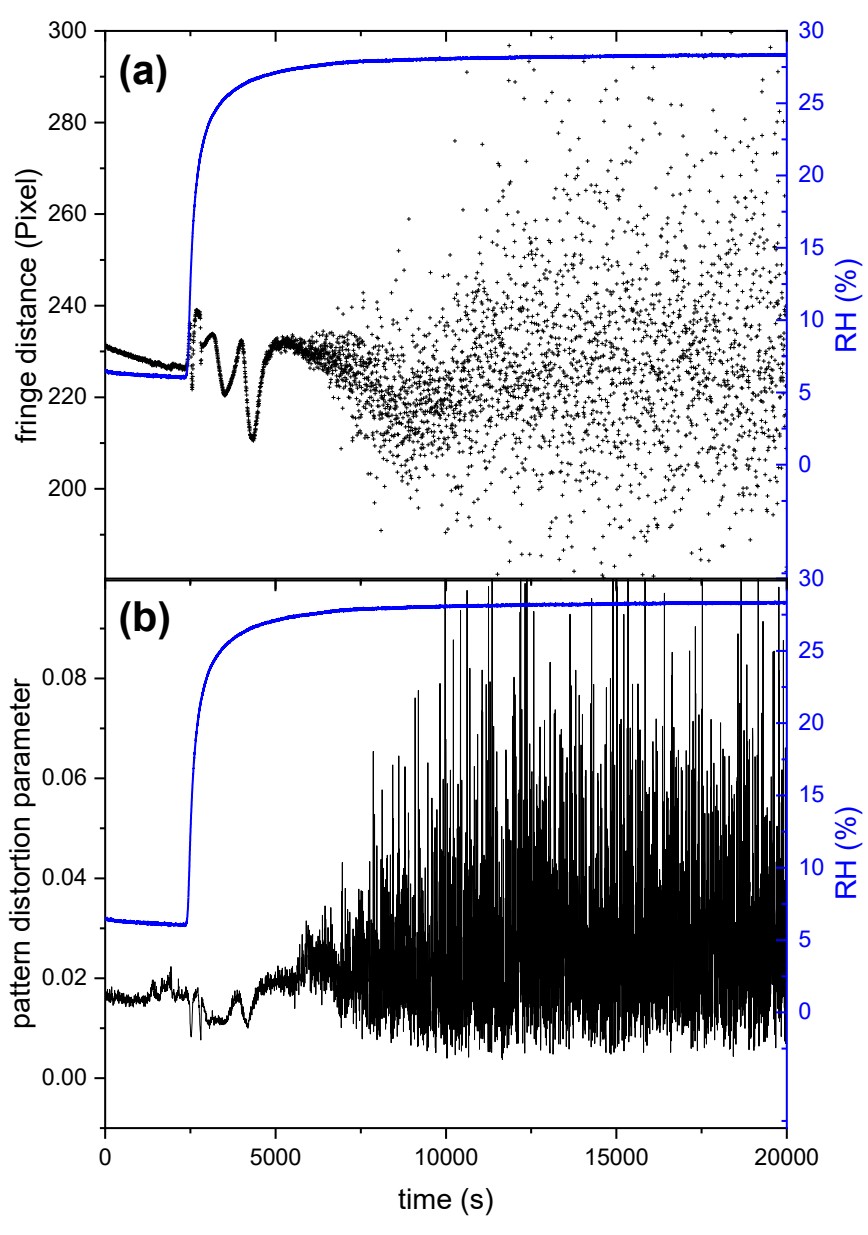



**Figure F-1: (a) Mean fringe distance and (b) pattern distortion parameter of AA particle exposed to UV and O$_3$ up to 26% mass loss upon increasing RH from 6.0% to 28.4%. Significant increase in fluctuations in fringe distance (a) and pattern distortion parameter (b) are observed at 6,000 s, indicating the efflorescence of AA.**

**Data availability.** All data of the figures are available online at https://doi.org/10.3929/ethz-c-000788184 (Antossian et al., 2025).

**Author contributions.** Funding was acquired by UKK. Experiments were conceptualized by UKK and CA and carried out by CA. The data were analyzed by CA and UKK. The manuscript was written by CA and revised by UKK and MM.

**Competing interests.** The authors declare that they have no conflict of interest.

**Acknowledgements.** The authors would like to thank Colette Heald for providing helpful feedback on the manuscript. The authors would like to acknowledge Kristopher McNeill and Oleksandr Yushckenko for their assistance in the absorbance measurements.

**Financial support.** This research has been supported by the Eidgenössische Technische Hochschule Zürich (grant no. 23-1 ETH-020)

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
