# Peer review of "Photochemical and ozone-induced aging significantly alter the viscosity of aqueous *trans*-aconitic acid aerosol particles"

_EGUsphere, 2025_

## Referee Comment (RC3)

**General comment:**

The manuscript by Antossian et al. uses trans-aconitic acid (AA) as a model system for atmospheric organic aerosol and investigates how different types of aging affect key particle properties, namely dynamic viscosity and hygroscopicity, using an electrodynamic balance (EDB) setup. As aging types, UV-aging is considered and compared to simultaneous aging of AA aerosols to UV light and ozone (O3). Such experiments considering the combined and possible synergistic effects of different aging agents provides a key advancement compared to previous studies that have often aged aerosols individually by either exposure to UV or O3 and investigated effects on particle properties. The authors demonstrate that aging led to significant enhancement of particle viscosity, while hygroscopicity decreased slightly. The results are interesting and the topic fits well within the scope of the journal. The manuscript is well written, and the arguments can be clearly followed. Below I list some specific comments that I encourage the authors to address in a revised version of their manuscript, to further clarify their observations.

**Specific comments:**

L50: Change to "O'Brien…". Also on e.g., L289, L302

L52: Formatting. Change to: "e.g. Kalberer et al (2004)."

L63: For condensed phase reactivity, consider adding (Kuwata and Martin, 2012)

L63: For effects on gas-particle partitioning, consider adding (Zaveri et al., 2014)

L70-72: Consider adding (Zelenyuk et al., 2012)

L79: Add: "low" relative humidities

L81: Consider adding something like: "… UV light and ozone, as simultaneously encountered in Earth's atmosphere during daytime."

L88-91 & L99: Further information should be provided how trans-aconitic acid was chosen as a model system for atmospheric SOA? Has this been identified in organic aerosol in the atmosphere? How does its light absorption compare to other atmospheric organics?

L99: Add "possible" synergistic effects …

L110: Change to "Milli-Q"

L111: Please indicate an approximate droplet size range used in your experiments in the text.

L116-118: Please elaborate how "the hygroscopic response of NaCl particles" were used to check on RH. Was the 1.5% uncertainty in RH determined from the NaCl experiments?

L141: Formatting of reference: "as introduced by Braun and Krieger (2001)". Also on e.g., L257, L302, L346, L380

L183: "yields"

L187: It is unclear to me what the "conventional additivity rule" is. Please clarify.

L189: Please use the same symbol for "kappa" throughout. Compare e.g., L189 and L190; L385, …

Fig. 5: If the efflorescence of AA is around 59.9% (L189), why do you still observe such as strong decrease in particle mass at RH below this ERH?

L209: "O'Meara"

L229: Consider adding (Chenyakin et al., 2017) as an example for where Rdiff > Rmatrix.

L315: Can you quantify the "slightly different particle sizes"?

L342: Please clarify what you mean with "partial wetting morphology"

L340-341: Can the authors provide further reasoning why the UV-illumination of AA did not result in changes in the TAOS pattern as compared to the ozonolysis experiments? I would expect UV aging to also cause formation of small, oxidized molecules that could undergo liquid-liquid phase separation from larger oligomers. On the other hand, you suggest that the O:C ratio before vs. after aging is similar (L435-451). Would this not go against your hypothesis of particles undergoing LLPS or am I not understanding your argument correctly?

L350: "Precipitation of trans-aconitic acid". Is this feasible? What is the deliquescence RH of (unaged/pure) AA? I.e., do you expect solid AA at the RH of your experiments (~78%)? Please provide further details. This would also help to understand your argumentation on L371.

Fig. 10: It could be nice to add curves for one of your UV experiments to illustrate the difference between both aging types.

L363: Consider citing (Marcolli et al., 2004)

L387: Consider citing (Petters and Kreidenweis, 2007)

Fig. 11: It could be helpful to cut off the y-axis at ~0.3 to improve readability of your experimental data. In addition, the comparison to ammonium sulfate is not discussed in the text.

L450: "Note that if…" This is an interesting thought. Could it be that the increase in O:C and increase in average molecular weight upon aging "balance" each other so that the hygroscopicity is largely unaffected (as suggested by your Fig. 11), whereas this would both promote an increase in viscosity (as suggested by your Fig. 12)?

L471: "presence of water typically slows down oligomerization". Please add appropriate reference.

L498: Please specify "dry"? Do you mean 17.7%? Also, please specify the temperature for which your mixing-time calculations were performed.

Section 4:

- I would like to see some further discussion on the atmospheric implications and relevance of your results. E.e., can you provide some thoughts on how your results affect the atmospheric processes that you motivate on L63-72? Can the characteristic mixing times (L494-498) be used to draw conclusions about e.g., PAH oxidation rates in the atmosphere?
- I would like to encourage the authors to comment on possible "limitations" of their experiments.
    - E.g., UV and or UV + O3 aging was done at high RH in this study. Can you provide some thoughts on how performing similar experiments at lower RH would affect your conclusion? Could O3 aging become kinetically limited at lower RH?
- To what extent does the relatively large particle size of several micrometer used in these experiments affect the results? I would expect O3 aging to be sensitive to particle surface-to-volume ratio, but UV-aging not.
- The brief outlook could be strengthened.
    - E.g., you would really benefit from measuring both the gas and particle phase composition during aging in future studies.
    - Lastly, the importance of future aging studies, similar to yours, that simultaneously consider different aging types could be further clarified.

Chenyakin, Y., Ullmann, D. A., Evoy, E., Renbaum-Wolff, L., Kamal, S., and Bertram, A. K.: Diffusion coefficients of organic molecules in sucrose–water solutions and comparison with Stokes–Einstein predictions, Atmos. Chem. Phys., 17, 2423–2435, https://doi.org/10.5194/acp-17-2423-2017, 2017.

Kuwata, M. and Martin, S. T.: Phase of atmospheric secondary organic material affects its reactivity, Proceedings of the National Academy of Sciences of the United States of America, 109, 17354–17359, https://doi.org/10.1073/pnas.1209071109, 2012.

Marcolli, C., Luo, B. P., and Peter, T.: Mixing of the organic aerosol fractions: Liquids as the thermodynamically stable phases, Journal of Physical Chemistry A, 108, 2216–2224, https://doi.org/10.1021/jp036080l, 2004.

Petters, M. D. and Kreidenweis, S. M.: A single parameter representation of hygroscopic growth and cloud condensation nucleus activity, Atmos. Chem. Phys., 7, 1961–1971, https://doi.org/10.5194/acp-7-1961-2007, 2007.

Zaveri, R. A., Easter, R. C., Shilling, J. E., and Seinfeld, J. H.: Modeling kinetic partitioning of secondary organic aerosol and size distribution dynamics: representing effects of volatility, phase state, and particle-phase reaction, Atmos. Chem. Phys., 14, 5153–5181, https://doi.org/10.5194/acp-14-5153-2014, 2014.

Zelenyuk, A., Imre, D., Beranek, J., Abramson, E., Wilson, J., and Shrivastava, M.: Synergy between Secondary Organic Aerosols and Long-Range Transport of Polycyclic Aromatic Hydrocarbons, Environ. Sci. Technol., 46, 12459–12466, https://doi.org/10.1021/es302743z, 2012.

---

## Author Comment (AC1)

The authors thank the Anonymous Referee #1 for the comments, which helps us to improve the manuscript.

*1) In Figure 2, it is not described from where the cross-section measurement is obtained.*

First, seven AA solutions having different concentrations (2.13 mol L$^{-1}$, 0.47 mol L$^{-1}$, 0.10 mol L$^{-1}$, $3.7 \times 10^{-2}$ mol L$^{-1}$, $8.7 \times 10^{-3}$ mol L$^{-1}$, $2.2 \times 10^{-3}$ mol L$^{-1}$, and $5.4 \times 10^{-4}$ mol L$^{-1}$) were prepared. Then, the absorbance of each solution was measured using the UV-Visible spectrophotometer. The cross section was then calculated using Beer-Lambert's law.

This description will be added in a revised version of the manuscript.

*2) For context, how does the light absorption of trans-aconitic acid compare with typical brown carbon chromophores, such as 4-nitrocatechol?*

We calculated the absorption cross section of 4-nitrocatechol based on the absorption spectrum given by (Cornard et al., 2005). The cross section of 4-nitrocatechol is about 3 orders of magnitude higher compared to *trans*-aconitic acid between 300 and 600 nm.

We will add a sentence comparing the two cross sections in a revised version of the manuscript to put the absorbance of AA into perspective.

*3) Are reactive nitrogen species formed during the production of ozone from air? Are these accounted for? Typically, pure oxygen is used to avoid the formation of additional reactive species.*

We use a low-pressure mercury lamp based generator similar to the one used by Birks et al. (Birks et al., 2018) for producing ozone using synthetic air (5.0) as a molecular oxygen source. Ozone is generated by photolysis of $O_2$ at the 184.9 nm line of the lamp. While some $O(^1D)$ is produced through photolysis of $O_3$ at the 253.7 nm line of mercury which could produce NO from $N_2$ in principle, it is quenched by reactions with $O_2$, water vapor and $N_2$, so that effectively the production of nitrogen species are negligible (Birks et al., 2018; Lambe et al., 2017).

*4) How exactly was the irradiance at the particle calculated? 0.16 W/cm2 seems low for a laser that has beam diameter of <1.5mm.*

We thank the reviewer very much for asking, as we indeed made a mistake in our analysis. The laser was set to a nominal power of 15.0 mW, which leads to a power of 10.6 mW (measured with an Ophir power meter) after reflections from two mirrors needed for beam steering in free space. In addition, the beam has to pass through a thin window which reflects and attenuates the power by a factor measured to be 0.92, i.e. the beam power at the particle was 9.8 mW. We measured the beam profile (Cinogy Nano, Germany) to have a cross section of $1.0 \cdot 10^{-2}$ cm$^2$ based on a Gaussian fit to the profile. This leads to an irradiance of (1.0+-0.3) W/cm$^2$, with the overall uncertainty (from power measurement and beam profile measurement) conservatively estimated to be 30%.

We will correct this error in a revised manuscript.

*5) To account for the Stokes force, the speed of the gas flow over the particle is needed. What is the speed of the gas and how was this determined?*

All experiments were carried out using a gas flow of 50 sccm. Thus, the DC voltage that needs to be applied to keep the particle levitated in the EDB, compensates for the Stokes force of the gas flow in addition to the gravitational force. To measure the drag force, the gas flow was set to 0 sccm and the DC voltage that needed to be applied to keep the particle levitated was recorded. Based on the difference between the DC voltage at 50 sccm and 0 sccm, the drag force was deduced for the specific size of the particle at the time of measurement. For the subsequent measurements the influence of size on drag force was taken into account using Stoke's law.

We will extend the experimental description accordingly in a revised manuscript.

*6) In calculating Dw, a large RH step change is used. Given that Dw changes significantly with RH, what RH does the inferred value of Dw correspond to?*

The inferred $D_w$ corresponds to a RH ~ 17%, which is the midpoint of the step in RH 6-28%, based on the assumption that the $a_w$ dependence of $D_w$ is linear on a log scale. The reviewer is correct that smaller RH steps would yield a more accurate diffusivity data, however in our experimental set up, we are limited by a resolution in mass change as well as in radius change and thus needed to have a sufficiently large change in both.

*7) The use of fractional Stokes-Einstein in single and multi-component particles has been explored by Sheldon et al. (https://doi.org/10.1039/D2EA00116K), with a focus on citric acid. For pure citric acid particle, the fSE works well. However, the addition of co-solutes changes the behavior significantly. The reacted mixtures studied here will likely not follow a straightforward fSE relationship either, although I think that without direct viscosity characterization, this is the only way to infer anything about viscosity. I suggest the authors include this reference with some discussion on the limitations of using fSE in multicomponent mixtures.*

We agree with the reviewer that inferring viscosity from water diffusivity measurements requires a number of assumptions, which need justification.

First, there is the general problem of estimating viscosities of multi-component mixtures. Even for aqueous ternary mixtures where viscosity data of the aqueous binary subsystems are available, there is not yet agreement which mixing rule should be applied, although Klein et al. (Klein et al., 2024) come to the conclusion that a simple mixing rule based on mole fractions performs well. However, this does not to hold true for systems forming a two-phase gel structure. These gel structures often form in electrolytes with divalent cations (e.g. (Richards et al., 2020); (Sheldon et al., 2023b), (Sheldon et al., 2023a)) but are not expected to occur in organic mixtures. Hence, with progressing chemical transformation we expect to form a homogeneous multi-component mixture.

Second, when applying fractional Stokes-Einstein (fSE), how is the fractional exponent changing with exposure? We thank the reviewer for raising this question. Generally, often we are not interested in the viscosity per se, but rather in the diffusivity of various species, e.g. in water diffusivity for hygroscopic response or oxygen diffusivity for chemical reactivity within the condensed phase. Viscosity serves in this context as a property which allows estimating the diffusivity of different species assuming an effective radius of the species of interest. Fractional Stokes-Einstein is just an empirical relationship for calculating diffusivity from viscosity (or the

other way round) based on the limited data which is presently available. In the manuscript, we assume a constant fractional exponent (that of citric acid) even though the composition of the matrix is changing considerably. However, if we look at the empirical relationship for the fractional exponent vs the ratio of the radii of the diffusing species ($R_{diff}$) over the effective radius of the organic matrix molecules ($R_{matrix}$), $R_{diff}/R_{matrix}$, (Evoy et al., 2020), Fig. 3, we see that the exponent becomes increasingly smaller with a decreasing ratio. Since our data indicate that smaller fragmentation products volatilize and are lost to the gas phase, we expect the ratio to decrease because of the increase in average $R_{matrix}$. Such increase leads to larger viscosities in fSE calculations upon estimating viscosity from water diffusivity data. Therefore, we conclude that the viscosity estimates presented in the preprint are lower limit viscosities.

We will add this discussion to a revised version of the manuscript.

*8) Is there any possibility that the particle charge changes over the course of oxidation due to the formation of charged species that are taken up or lost by the particles?*

Charge separation by volatilization is not expected to happen due to strong Coulomb interaction between the charged molecules.

*9) Corresponding to Figure 8, the decay with ozone is described as linear. However, it is not clear to me that this is not simply a slow exponential decay. The inference of a reaction order from these data is also not clear to me, due to the coupling of particle mass with the volatility of the products. The data does not show the change in the concentration of the reactant with time and, thus, may be of limited use when determining reaction kinetics.*

We agree with the Anonymous Referee #1 that the order of the reaction cannot be inferred directly from our data of the production of volatile products and we are thankful for pointing this error. This wrong statement will be corrected in this revised version of the manuscript.

*10) How were diffusivity and viscosity estimated made on unreacted particles that exhibited efflorescence?*

Most of the unreacted AA particles effloresced at RH ~59.9%. However, some of the unreacted AA particles did not exhibit any efflorescence upon drying. Unreacted AA particles which did not effloresce were used to determine the diffusivity of unreacted AA using the same technique used for the reacted particles.

*Other issues:*

*The general structure of the article is a little unusual, with results appear in the methods section. This creates a slightly awkward flow to the manuscript.*

Some results were shown in the methods section to facilitate the reader in understanding the methods. Figures 5 and 6 were added in the methods section to show an example of how hygroscopicity and diffusivity measurements were performed. Figure 2 was added in the materials section because the purpose is to justify the choice of AA a surrogate for SOA and thus it is not considered to be one of the important results of this manuscript.

Birks, J. W., Williford, C. J., Andersen, P. C., Turnipseed, A. A., Strunk, S., and Ennis, C. A.: Portable ozone calibration source independent of changes in temperature, pressure and humidity for research and regulatory applications, Atmos. Meas. Tech., 11, 4797-4807, doi: 10.5194/amt-11-4797-2018, 2018.

Cornard, J. P., Rasmiwetti, and Merlin, J. C.: Molecular structure and spectroscopic properties of 4-nitrocatechol at different pH: UV-visible, Raman, DFT and TD-DFT calculations, Chem. Phys., 309, 239-249, 10.1016/j.chemphys.2004.09.020, 2005.

Evoy, E., Kamal, S., Patey, G. N., Martin, S. T., and Bertram, A. K.: Unified Description of Diffusion Coefficients from Small to Large Molecules in Organic-Water Mixtures, J. Phys. Chem. A, 124, 2301-2308, doi:10.1021/acs.jpca.9b11271, 2020.

Klein, L. K., Bertram, A. K., Zuend, A., Gregson, F., and Krieger, U. K.: Viscosity of aqueous ammonium nitrate-organic particles: equilibrium partitioning may be a reasonable assumption for most tropospheric conditions, Atmos. Chem. Phys., 24, 13341-13359, 10.5194/acp-24-13341-2024, 2024.

Lambe, A., Massoli, P., Zhang, X., Canagaratna, M., Nowak, J., Daube, C., Yan, C., Nie, W., Onasch, T., Jayne, J., Kolb, C., Davidovits, P., Worsnop, D., and Brune, W.: Controlled nitric oxide production via O(1D) + N2O reactions for use in oxidation flow reactor studies, Atmos. Meas. Tech., 10, 2283-2298, doi: 10.5194/amt-10-2283-2017, 2017.

Richards, D. S., Trobaugh, K. L., Hajek-Herrera, J., Price, C. L., Sheldon, C. S., Davies, J. F., and Davis, R. D.: Ion-molecule interactions enable unexpected phase transitions in organic-inorganic aerosol, Sci. Adv., 6, 11, 10.1126/sciadv.abb5643, 2020.

Sheldon, C. S., Choczynski, J. M., Morton, K., Diaz, T. P., Davis, R. D., and Davies, J. F.: Exploring the hygroscopicity, water diffusivity, and viscosity of organic-inorganic aerosols - a case study on internally-mixed citric acid and ammonium sulfate particles, Environ. Sci. - Atmospheres, 3, 24-34, 10.1039/d2ea00116k, 2023a.

Sheldon, C. S., Salazar, J., Diaz, T. P., Morton, K., Davis, R. D., and Davies, J. F.: Correlating the Viscosity and Rate of Water Diffusion in Semisolid Gel-Forming Aerosol Particles, ACS Earth Space Chem., 7, 2311-2320, 10.1021/acsearthspacechem.3c00241, 2023b.

---

## Author Comment (AC2)

The authors thank the Anonymous Referee #2 for the positive review and the comments, which will help us to improve the manuscript.

*Efflorescence point of untreated AA - You report that untreated AA typically effloresced at RH 59.9%. Does this value fall within the known range of values in the literature, or at least beneath any reported deliquescence points? And if so, can reference(s) be provided?*

To the best of our knowledge, the deliquescence or efflorescence of *trans*-aconitic acid is not reported in the literature. However, for *cis*-aconitic acid, one study reported that no crystallization was observed (Han et al., 2022). However, also in our experiments, some of the unreacted particles did not exhibit any efflorescence as well upon drying.

This information will be added to the beginning of section 3.4 in the revised manuscript.

*Assumption regarding κ retrieval after aging - The statement that fitting at higher RH provides a better estimate of κ, as deviations from ideality become less important with dilution, is reasonable. However, this implicitly assumes that no further chemical processing occurs once UV and $O_3$ are switched off. It would be useful to state this assumption explicitly. It would also be good to discuss whether the possibility of slow processing post UV and/or ozonolysis is (un)likely.*

The assumption that no further chemical processing occurs once UV and $O_3$ are switched off will be added in a revised version of the manuscript.

In the example shown in Fig. 4, we do not observe significant mass loss after switching off UV and $O_3$, once the RH and temperature are stabilized. Thus, it is unlikely that the reaction continues upon drying. In addition, since the viscosity of the particles is higher after chemical processing, it would be more difficult for the particles to undergo any reaction upon decreasing the RH. However, as shown in Fig. 6, slight mass loss is observed upon increasing the RH from 6.3% to 28.2% and keeping the particle at elevated RH for several hours. The observed mass loss could be due to volatilization of some of the remaining products or slow chemical processing. This mass loss is more significant after the second RH step (28.2% to 49.4%), which might indicate further chemical processing at higher RH. The discussion will be added in a revised version of the manuscript.

*Kinetic limitations - related to the previous point, the κ interpretation also appears to assume no kinetic limitations to hygroscopic mass transfer. It may be worth briefly commenting on whether slow diffusion / kinetic constraints could influence κ retrieval, particularly given the wider viscosity discussion.*

For κ retrieval, we use the RH range 62%-80%. At elevated RH, we do not expect any kinetic constraints that could influence κ retrieval. This can be seen also in Fig. 4. While the first RH step (6.3% to 28.2%) shows kinetic limitations, the second RH step (28.2% to 49.4%) does not show any limitations. This indicates that even particles that become viscous at low RH, at RH above 50%, the water diffusivity is greater than $2 \times 10^{-9}$ $cm^2 s^{-1}$, see discussion in Appendix D. This ensures that the particles in our set-up are close to thermodynamic equilibrium in the range where we measure κ.

*Link to brown carbon potential - the results could be relevant to the developing brown carbon narrative. You already hint at the formation of potentially more photoreactive intermediates when describing the non-linear photolysis kinetics ("slow in the beginning, then speeds up..."). Some speculation on the possible implications for BrC formation or optical properties could strengthen the wider atmospheric relevance of the work.*

We thank the reviewer for pointing out this possible implication. We will extend the third paragraph of section 3.2 accordingly:

*"In contrast to ozonolysis, the shape of the photolysis decay kinetics is not linear. It is slow in the beginning, then speeds up possibly due to the formation of more photoreactive intermediates or due to the initiation of an autocatalytic process. After about 60% mass loss, the reaction slows down again, most likely because most photoreactive species are consumed and non-absorbing products are formed (photobleaching) or because of radical recombination. We conclude that similar to the findings by (O'brien and Kroll, 2019) and (Sun and Smith, 2024), initial photolytic mass loss cannot be extrapolated to the entire SOA mass loss and that a photo-recalcitrant fraction remains, which prevents or slows down further mass loss. In the study by Sun and Smith (2024), Suwanee River Fulvic Acid (SRFA) was used as a surrogate for brown carbon SOA. They observed that UV photolysis of an aqueous SRFA solution results in photobleaching at UV wavelengths and enhanced absorbance (photoenhancement) at visible wavelengths (Sun and Smith, 2024). For AA, UV aging most likely results initially in photoenhancement. As the reaction proceeds, photobleaching occurs due to degradation of chromophores during fragmentation, resulting in a photo-recalcitrant fraction. These findings suggest that even a small precursor molecule, such as AA, might demonstrate a behavior similar to atmospheric brown carbon, exhibiting an alteration of optical properties upon UV aging. However, further investigations are needed to prove that this mechanism is responsible for the observed kinetics."*

*I think in the text and references "O'meara" should be "O'Meara"*

This error will be corrected in a revised manuscript.

References

Han, S., Hong, J., Luo, Q. W., Xu, H. B., Tan, H. B., Wang, Q. Q., Tao, J. C., Zhou, Y. Q., Peng, L., He, Y., Shi, J. N., Ma, N., Cheng, Y. F., and Su, H.: Hygroscopicity of organic compounds as a function of organic functionality, water solubility, molecular weight, and oxidation level, Atmos. Chem. Phys., 22, 3985-4004, doi:10.5194/acp-22-3985-2022, 2022.
O'Brien, R. E. and Kroll, J. H.: Photolytic Aging of Secondary Organic Aerosol: Evidence for a Substantial Photo-Recalcitrant Fraction, J. Phys. Chem. Lett., 10, 4003-4009, doi:10.1021/acs.jpclett.9b01417, 2019.
Sun, M. R. and Smith, G. D.: Photolytic Mass Loss of Humic Substances Measured with a Quartz Crystal Microbalance, ACS Earth Space Chem., 8, 1623-1633, doi:10.1021/acsearthspacechem.4c00134, 2024.

---

## Author Comment (AC3)

The authors thank the Anonymous Referee #3 for the comments, which will help us to improve the manuscript.

**Specific comments:**

*L50: Change to "O'Brien...". Also on e.g., L289, L302*

This error will be corrected in a revised manuscript.

*L52: Formatting. Change to: "e.g. Kalberer et al (2004)."*

This error will be corrected in a revised manuscript.

*L63: For condensed phase reactivity, consider adding (Kuwata and Martin, 2012)*

This reference will be added in a revised manuscript.

*L63: For effects on gas-particle partitioning, consider adding (Zaveri et al., 2014)*

This reference will be added in a revised manuscript.

*L70-72: Consider adding (Zelenyuk et al., 2012)*

This reference will be added in a revised manuscript.

*L79: Add: "low" relative humidities*

We thank the reviewer for pointing this out. "low" relative humidities will be added in a revised manuscript.

*L81: Consider adding something like: "... UV light and ozone, as simultaneously encountered in Earth's atmosphere during daytime."*

This phrase will be added to the sentence in line 81 in a revised manuscript.

*L88-91 & L99: Further information should be provided how trans-aconitic acid was chosen as a model system for atmospheric SOA? Has this been identified in organic aerosol in the atmosphere? How does its light absorption compare to other atmospheric organics?*

*Trans*-aconitic acid was chosen because of its favorable physicochemical properties and not because of its presence in the atmosphere. Due to the constraints of our experimental set-up, only limited number of organic compounds can be studied. First, the organic compound must have a carbon-carbon double bond to react with $O_3$. Second, it must be water soluble and must not crystallize easily. It is important to have a liquid droplet to enable the measurement of its

size by analyzing its TAOS pattern. Third, it must have low vapor pressure so it can stay levitated in the EDB for a long time without significant evaporation. Non-volatile compounds allow a more precise measurement of size because the change in size of the particle would be attributed solely to aging and not to loss of the pure compound by evaporation. Fourth, it must have low viscosity to prevent mass transfer limitations before aging occurs. Fifth, it must have a high reaction rate coefficient with $O_3$ to allow significant aging in reasonable experimental time spans. Finally, it must photolyze either directly at wavelengths longer than 300 nm or indirectly by using a photosensitizer.

Several acids were tested including shikimic acid, maleic acid, citraconic acid, and *trans*-aconitic acid. The reaction of shikimic acid with $O_3$ and UV was very slow, maleic acid crystallized very quickly, while citraconic acid was volatile. Since *trans*-aconitic acid fulfilled most of the mentioned criteria, it was chosen as a proxy system.

For the question concerning the absorption of AA, we refer to our answer to a related question by reviewer #1:

"We calculated the absorption cross section of 4-nitrocatechol based on the absorption spectrum given by (Cornard et al., 2005). The cross section of 4-nitrocatechol is about 3 orders of magnitude higher compared to trans-aconitic acid between 300 and 600 nm.

We will add a sentence comparing the two cross sections in a revised version of the manuscript to put the absorbance of AA into perspective."

*L99: Add "possible" synergistic effects ...*

"possible" will be added in a revised manuscript.

*L110: Change to "Milli-Q"*

"Milli-Q" will be corrected in a revised manuscript.

*L111: Please indicate an approximate droplet size range used in your experiments in the text.*

According to the technical specification of the inkjet cartridge, the size of the ejected droplets is ~37 µm. We start with a 4 wt-% AA solution with water content equilibrating quickly to RH in the environmental cell. Depending on RH, the size range in the experiments is from ~6 µm under dry conditions after mass loss due to reaction to ~20 µm at high RH before reaction.

We will add size range in a revised manuscript.

*L116-118: Please elaborate how "the hygroscopic response of NaCl particles" were used to check on RH. Was the 1.5% uncertainty in RH determined from the NaCl experiments?*

The humidity sensor used (SHT85) is factory calibrated to ± 1.5% accuracy at the temperatures and relative humidities of our experiments. We checked this by measuring the deliquescence RH (DRH) of NaCl, which is well established in the literature. We found agreement with the calibration of the manufacturer.

We will modify the sentence in line 116 to make clear that these were deliquescence measurements.

L141: Formatting of reference: "as introduced by Braun and Krieger (2001)". Also on e.g., L257, L302, L346, L380

This reference will be formatted in a revised manuscript.

*L183: "yields"*

This grammatical mistake will be corrected in a revised manuscript.

*L187: It is unclear to me what the "conventional additivity rule" is. Please clarify.*

The additivity rule used is mass fraction based and can be found in Appendix B:

$$\rho(w_s) = \left( \frac{1-w_s}{\rho_w} + \frac{w_s}{\rho_0} \right)$$

where $\rho(w_s)$ is the density of solute at mass fraction $w_s$, $\rho_w$ is the density of water, we retrieve $\rho_0$, the density of aqueous AA solute at dry conditions, i.e. the subcooled melt density.

More details on the choice of this equation can found in (Lienhard et al., 2012).

*L189: Please use the same symbol for "kappa" throughout. Compare e.g., L189 and L190; L385, …*

Same symbol for "kappa" will be used in a revised manuscript.

*Fig. 5: If the efflorescence of AA is around 59.9% (L189), why do you still observe such as strong decrease in particle mass at RH below this ERH?*

The efflorescence of uncreated AA particle is typically around 59.9%. However, this figure corresponds to AA particle, which has been reacted with UV and $O_3$. AA particles after UV and $O_3$ exposure did not exhibit any efflorescence upon drying, as mentioned in section 3.4.

*L209: "O'Meara"*

This error will be corrected in a revised manuscript.

*L229: Consider adding (Chenyakin et al., 2017) as an example for where Rdiff > Rmatrix.*

This reference will be added in a revised manuscript.

*L315: Can you quantify the "slightly different particle sizes"?*

The size of particle 1 at the beginning of the reaction is 14.6 μm, while that of particle 2 is 10.4 μm.

L342: Please clarify what you mean with "partial wetting morphology"

LLPS can exhibit different morphologies, such as core-shell and partial wetting. Core-shell occurs when one phase of the aerosol particle is totally wetted by the other, while partial wetting occurs when one phase is partially engulfed by the other, thus creating asymmetric structures (Qiu and Molinero, 2015). Schematics of different morphologies are illustrated in (Qiu and Molinero, 2015).

If LLSP occurs with core-shell morphology, the TAOS pattern remains that of spherical symmetric particle, while with partial wetting, it becomes irregular. For this reason, we suggest that if LLPS occurred, it would have partial wetting morphology.

*L340-341: Can the authors provide further reasoning why the UV-illumination of AA did not result in changes in the TAOS pattern as compared to the ozonolysis experiments? I would expect UV aging to also cause formation of small, oxidized molecules that could undergo liquid-liquid phase separation from larger oligomers.*

It is still possible that liquid-liquid phase separation occurred upon UV reaction, but if it happened with core-shell morphology, then the TAOS pattern will remain that of a spherical symmetric particle, thus we cannot observe the phase separation.

*On the other hand, you suggest that the O:C ratio before vs. after aging is similar (L435-451). Would this not go against your hypothesis of particles undergoing LLPS or am I not understanding your argument correctly?*

This section refers to particles exposed to UV irradiation or UV and $O_3$ simultaneously and not to particles exposed to $O_3$ alone. Because of the irregularity in the TAOS pattern of the particles exposed to $O_3$ alone, hygroscopicity measurements were not carried out on the particle aged with $O_3$ alone, thus we cannot deduce whether O:C ratio changed after ozonolysis.

*L350: "Precipitation of trans-aconitic acid". Is this feasible? What is the deliquescence RH of (unaged/pure) AA? I.e., do you expect solid AA at the RH of your experiments (~78%)? Please provide further details. This would also help to understand your argumentation on L371.*

The deliquescence RH of pure AA based on our experiments is about 96%. All AA particles at the beginning of the experiments are liquid, since the reactions are carried out at RH above 78%, above the highest efflorescence RH observed in our experiments (~59.9%). This can be seen clearly from the TAOS pattern, which is spherical, unlike solid particles, which exhibit irregular TAOS pattern.

If precipitation of AA occurs after an hour of ozone exposure, part of the particle could be crystallized AA within a multicomponent liquid mixture.

*Fig. 10: It could be nice to add curves for one of your UV experiments to illustrate the difference between both aging types.*

We thank the reviewer for the suggestion. Figure 10 will be modified to include an example of UV aging.

*L363: Consider citing (Marcolli et al., 2004)*

This reference will be added in a revised manuscript.

*L387: Consider citing (Petters and Kreidenweis, 2007)*

This reference will be added in a revised manuscript.

*Fig. 11: It could be helpful to cut off the y-axis at ~0.3 to improve readability of your experimental data. In addition, the comparison to ammonium sulfate is not discussed in the text.*

Putting ammonium sulfate as a reference illustrates for the reader that organics have a significantly lower κ. We will change the first two sentences starting in line 385 to:

*"Figure 11 shows the hygroscopicity results in terms of derived κ values. For reference, a selection of κ values of other organic acids and ammonium sulfate are taken from the literature. As shown in this figure, organic acids can have quite different κ values, but are significantly lower than inorganic salts."*

*L450: "Note that if…" This is an interesting thought. Could it be that the increase in O:C and increase in average molecular weight upon aging "balance" each other so that the hygroscopicity is largely unaffected (as suggested by your Fig. 11), whereas this would both promote an increase in viscosity (as suggested by your Fig. 12)?*

Yes, exactly, this is what we meant by this sentence.

*L471: "presence of water typically slows down oligomerization". Please add appropriate reference.*

(Kalberer et al., 2004)

*L498: Please specify "dry"? Do you mean 17.7%? Also, please specify the temperature for which your mixing time calculations were performed.*

Our calculations were done for the viscosity at the RH of our experiments (~17%) and at room temperature (~298 K). We know from our experiments that at RH 38%, the viscosity is already significantly lower. This would imply that at 298 K, the boundary layer RH needs to be below 20% to yield these long mixing times. However, since lower temperature increases the viscosity, in a colder boundary layer, higher RH may lead to similar mixing times.

We will add the RH and temperature to line 497.

**Section 4:**

*- I would like to see some further discussion on the atmospheric implications and relevance of your results. E.e., can you provide some thoughts on how your results affect the atmospheric processes that you motivate on L63-72? Can the characteristic mixing times (L494-498) be used to draw conclusions about e.g., PAH oxidation rates in the atmosphere?*

Since the characteristic mixing time increases after aging, we expect longer-range transport of the remaining particle. However, this is not directly related to the oxidation rate of PAHs in the atmosphere.

In addition, the lifetime of the particle in the atmosphere increases because it becomes less susceptible to washout and further chemical processing as the viscosity increases.

We will add one sentence about the potential implications of the characteristic mixing times on aerosol lifetime and long-range transport to line 498.

*- I would like to encourage the authors to comment on possible "limitations" of their experiments.*

    ○   *E.g., UV and or UV + $O_3$ aging was done at high RH in this study. Can you provide some*

*thoughts on how performing similar experiments at lower RH would affect your conclusion?*

*Could $O_3$ aging become kinetically limited at lower RH?*

Since most unreacted AA particles effloresced at ~59.9% RH, it is not possible to perform the aging experiments at lower RH. This is because we need the TAOS pattern to measure the change in size and then deduce the viscosity. Since solid AA particles exhibit irregular TAOS pattern, quantitative analysis would be difficult.

$O_3$ aging could become kinetically limited at lower RH, based on previous studies (Berkemeier et al., 2016; Steimer et al., 2014; Kohli et al., 2024). For instance, in the case of ozonolysis of shikimic acid particles, the bulk diffusion coefficients of both ozone and shikimic acid increased by several orders of magnitude as RH increased, since the particle phase changed from amorphous solid to semisolid and liquid, suggesting that the reaction occurs at the near particle surface at low RH and is kinetically limited by slow diffusion (Berkemeier et al., 2016).

From our point of view, the biggest limitation of our study is the missing chemical information during aging, which makes it difficult to understand the differences that we observe. This limitation is already mentioned throughout the text.

*- To what extent does the relatively large particle size of several micrometer used in these experiments affect the results? I would expect $O_3$ aging to be sensitive to particle surface-to-volume ratio, but UV-aging not.*

We agree with the reviewer that ozone aging is sensitive to particle surface-to-volume ratio, while UV aging is not. We expect the rate of ozonolysis to increase with smaller particles, while the rate of UV reaction is less sensitive. However, here we are not providing any rate dependence

on ozone concentration or UV intensity. We are only interested in the different properties after aging, which are independent of size because the mechanism should be similar.

*- The brief outlook could be strengthened.*

- *E.g., you would really benefit from measuring both the gas and particle phase composition during aging in future studies.*
- *Lastly, the importance of future aging studies, similar to yours, that simultaneously consider different aging types could be further clarified.*

Concentrations of products in gas phase emitted in single particle experiments are so low that we cannot measure them with current technology. However, flow tube or chamber experiments could be used.

We will elaborate the outlook according to these suggestions in a revised manuscript.

References

Berkemeier, T., Steimer, S. S., Krieger, U. K., Peter, T., Pöschl, U., Ammann, M., and Shiraiwa, M.: Ozone uptake on glassy, semi-solid and liquid organic matter and the role of reactive oxygen intermediates in atmospheric aerosol chemistry, Phys. Chem. Chem. Phys., 18, 12662-12674, doi:10.1039/c6cp00634e, 2016.
Chenyakin, Y., Ullmann, D. A., Evoy, E., Renbaum-Wolff, L., Kamal, S., and Bertram, A. K.: Diffusion coefficients of organic molecules in sucrose-water solutions and comparison with Stokes-Einstein predictions, Atmos. Chem. Phys., 17, 2423-2435, doi:10.5194/acp-17-2423-2017, 2017.
Cornard, J. P., Rasmiwetti, and Merlin, J. C.: Molecular structure and spectroscopic properties of 4-nitrocatechol at different pH: UV-visible, Raman, DFT and TD-DFT calculations, Chem. Phys., 309, 239-249, doi: 10.1016/j.chemphys.2004.09.020, 2005.
Kalberer, M., Paulsen, D., Sax, M., Steinbacher, M., Dommen, J., Prevot, A. S. H., Fisseha, R., Weingartner, E., Frankevich, V., Zenobi, R., and Baltensperger, U.: Identification of polymers as major components of atmospheric organic aerosols, Science, 303, 1659-1662, doi:10.1126/science.1092185, 2004.
Kohli, R. K., Reynolds, R. S., Wilson, K. R., and Davies, J. F.: Exploring the influence of particle phase in the ozonolysis of oleic and elaidic acid, Aerosol Sci. Technol., 58, 356-373, doi:10.1080/02786826.2023.2226183, 2024.
Kuwata, M. and Martin, S. T.: Phase of atmospheric secondary organic material affects its reactivity, Proc. Natl. Acad. Sci. U. S. A., 109, 17354-17359, doi:10.1073/pnas.1209071109, 2012.
Lienhard, D. M., Bones, D. L., Zuend, A., Krieger, U. K., Reid, J. P., and Peter, T.: Measurements of Thermodynamic and Optical Properties of Selected Aqueous Organic and Organic-Inorganic Mixtures of Atmospheric Relevance, J. Phys. Chem. A, 116, 9954-9968, doi:10.1021/jp3055872, 2012.
Marcolli, C., Luo, B. P., Peter, T., and Wienhold, F. G.: Internal mixing of the organic aerosol by gas phase diffusion of semivolatile organic compounds, Atmos. Chem. Phys., 4, 2593-2599, doi:10.5194/acp-4-2593-2004, 2004.

Petters, M. D. and Kreidenweis, S. M.: A single parameter representation of hygroscopic growth and cloud condensation nucleus activity, Atmos. Chem. Phys., 7, 1961-1971, doi:10.5194/acp-7-1961-2007, 2007.

Qiu, Y. Q. and Molinero, V.: Morphology of Liquid-Liquid Phase Separated Aerosols, J. Am. Chem. Soc., 137, 10642-10651, doi:10.1021/jacs.5b05579, 2015.

Steimer, S. S., Lampimäki, M., Coz, E., Grzinic, G., and Ammann, M.: The influence of physical state on shikimic acid ozonolysis: a case for in situ microspectroscopy, Atmos. Chem. Phys., 14, 10761-10772, doi:10.5194/acp-14-10761-2014, 2014.

Zaveri, R. A., Easter, R. C., Shilling, J. E., and Seinfeld, J. H.: Modeling kinetic partitioning of secondary organic aerosol and size distribution dynamics: representing effects of volatility, phase state, and particle-phase reaction, Atmos. Chem. Phys., 14, 5153-5181, doi:10.5194/acp-14-5153-2014, 2014.

Zelenyuk, A., Imre, D., Beránek, J., Abramson, E., Wilson, J., and Shrivastava, M.: Synergy between Secondary Organic Aerosols and Long-Range Transport of Polycyclic Aromatic Hydrocarbons, Environ. Sci. Technol., 46, 12459-12466, doi:10.1021/es302743z, 2012.